# The molecular and functional landscape of resistance to immune checkpoint blockade in melanoma

Su Yin Lim[1,2,14], Elena Shklovskaya [1,2,14], Jenny H. Lee[1,2,3], Bernadette Pedersen[1,2], Ashleigh Stewart[1,2], Zizhen Ming[1,2], Mal Irvine[1,2], Brindha Shivalingam[2,4,5], Robyn P. M. Saw [2,6,7], Alexander M. Menzies[2,7,8,9], Matteo S. Carlino[2,7,10,11], Richard A. Scolyer [2,7,12,13], Georgina V. Long [2,7,8,9,13] & Helen Rizos [1,2] ✉

Resistance to immune checkpoint inhibitor therapies in melanoma is common and remains an intractable clinical challenge. In this study, we comprehensively profile immune checkpoint inhibitor resistance mechanisms in short-term tumor cell lines and matched tumor samples from melanoma patients progressing on immune checkpoint inhibitors. Combining genome, transcriptome, and high dimensional flow cytometric profiling with functional analysis, we identify three distinct programs of immunotherapy resistance. Here we show that resistance programs include (1) the loss of wild-type antigen expression, resulting from tumor-intrinsic IFNγ signaling and melanoma de-differentiation, (2) the disruption of antigen presentation via multiple independent mechanisms affecting MHC expression, and (3) immune cell exclusion associated with PTEN loss. The dominant role of compromised antigen production and presentation in melanoma resistance to immune checkpoint inhibition highlights the importance of treatment salvage strategies aimed at the restoration of MHC expression, stimulation of innate immunity, and re-expression of wild-type differentiation antigens.

Immune checkpoint inhibitors targeting programmed cell death protein 1 (PD1) have revolutionized the treatment of metastatic melanoma patients, with phase III clinical trials (e.g., CheckMate-067, KEYNOTE-006) reporting objective response rates of 42–45% with an overall survival rate of 42% at 6.5 years[1–3]. Response rates and duration are further extended when PD1 inhibitors are combined with the cytotoxic T-lymphocyte-associated antigen 4 (CTLA4) inhibitor, ipilimumab (reviewed in ref. [4]). Nevertheless, resistance to immune checkpoint inhibitor therapies is common, with approximately 55% of melanoma patients showing innate resistance to single agent PD1 inhibitor (with

[1]Macquarie Medical School, Faculty of Medicine, Health and Human Sciences, Macquarie University, Sydney, NSW, Australia. [2]Melanoma Institute Australia, The University of Sydney, Sydney, NSW, Australia. [3]Department of Medical Oncology, Chris O'Brien Lifehouse, Sydney, NSW, Australia. [4]Department of Neurosurgery, Chris O'Brien Lifehouse, Sydney, NSW, Australia. [5]Department of Neurosurgery, Royal Prince Alfred Hospital, Sydney, NSW, Australia. [6]Department of Melanoma and Surgical Oncology, Royal Prince Alfred Hospital, Sydney, NSW, Australia. [7]Faculty of Medicine and Health, The University of Sydney, Sydney, NSW, Australia. [8]Department of Medical Oncology, Northern Sydney Cancer Centre, Royal North Shore Hospital, Sydney, NSW, Australia. [9]Department of Medical Oncology, Mater Hospital, Sydney, NSW, Australia. [10]Department of Medical Oncology, Blacktown Cancer and Haematology Centre, Blacktown Hospital, Sydney, NSW, Australia. [11]Department of Medical Oncology, Crown Princess Mary Cancer Centre, Westmead Hospital, Sydney, NSW, Australia. [12]Tissue Pathology and Diagnostic Oncology, Royal Prince Alfred Hospital and NSW Health Pathology, Sydney, NSW, Australia. [13]Charles Perkins Centre, The University of Sydney, Sydney, NSW, Australia. [14]These authors contributed equally: Su Yin Lim, Elena Shklovskaya. ✉e-mail: helen.rizos@mq.edu.au

~40% innate resistance to CTLA4+PD1 inhibitor combination), and almost 25% of responding patients acquiring resistance to PD1 inhibitor within 2 years of treatment[5].

Mechanisms associated with immune checkpoint inhibitor resistance have been reported in a small subset of patients, although no single resistance effector is strongly associated with resistance[6]. The most common drivers of resistance affect antigen presentation and are predicted to reduce tumor immunogenicity (reviewed in ref. [7]). These alterations include loss-of-function mutations in the beta-2-microglobulin (*B2M*) gene[8,9], downregulation of major

histocompatibility complex (MHC)-I expression[10,11], loss of MHC heterozygosity[12], and silencing of melanosomal wild-type differentiation antigens (e.g., MART-1/Melan-A, gp100, TYR) during melanoma de-differentiation[9,13–15]. Defects in IFNγ signaling (which also prevent IFNγ-induced MHC expression) have been identified in 4–10% of PD1-resistant melanomas and are often due to loss-of-function mutations in the IFNγ effector kinases JAK1 and JAK2[9,16].

Recent studies have identified features within the tumor microenvironment and the gut microbiome that are associated with immune checkpoint inhibitor resistance, but the precise

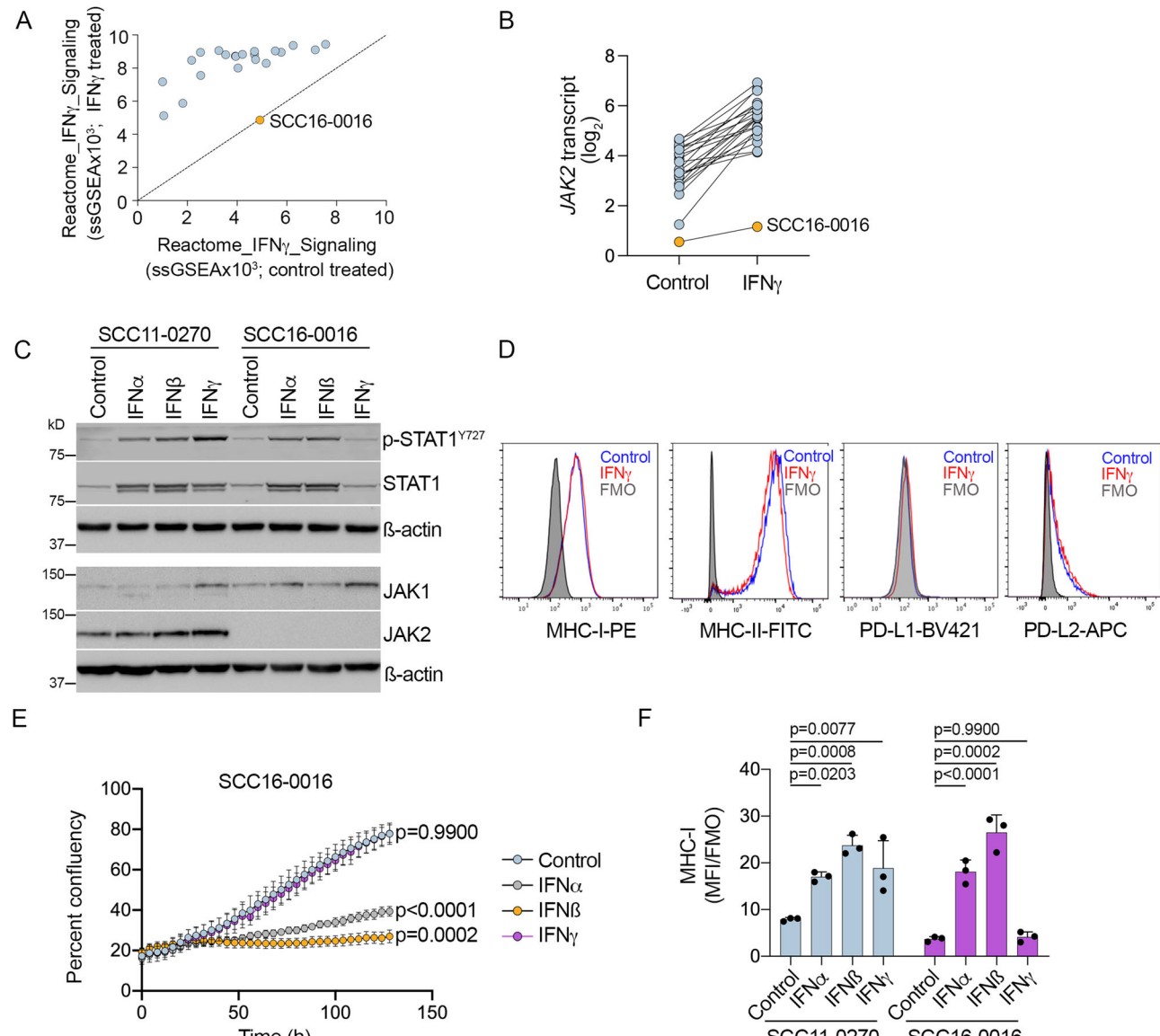

**Fig. 1 | IFNγ response in PD1 PROG cell lines. A** ssGSEA scores for the Reactome_Interferon_Gamma_Signaling gene set in 22 PD1 PROG melanoma cell lines treated with BSA control (*x*-axis) or 1000 U/ml IFNγ (*y*-axis) for 24 h. The SCC16-0016 PD1 PROG did not respond to IFNγ stimulation and is highlighted. **B** *JAK2* transcript expression in matched BSA control and IFNγ-treated PD1 PROG cell lines (*n* = 22). **C** Accumulation of phosphorylated STAT1 (p-STAT1$^{Y727}$), STAT1, JAK1, and JAK2 in the SCC11-0270 (responded to IFNγ) and SCC16-0016 (did not respond to IFNγ) PD1 PROG cell lines treated with BSA control, IFNα, IFNβ, or IFNγ (all at 1000 U/ml) for 24 h. **D** Representative histograms showing levels of MHC-I, MHC-II, PD-L1, and PD-L2 expression in SCC16-0016 cell line after BSA control (blue) or 1000 U/ ml IFNγ (red) treatment for 72 h. Fluorescence minus one (FMO) controls are shown as shaded histograms. **E** Representative proliferation curve of SCC16-0016 cells,

measured as percent confluence every 4 h, for up to 120 h, after treatment with BSA control, IFNα, IFNβ, or IFNγ (all at 1000 U/ml). Data shown are mean ± sd (six images per treatment per time point). Area under the curve data of biological triplicates were compared using one-way ANOVA with Dunnett correction for multiple comparisons, and adjusted *p*-values are shown. **F** Cell surface expression (median fluorescence intensity stained divided by fluorescence minus one control, MFI/FMO) of MHC-I in SCC11-0270 and SCC16-0016 PD1 PROG cell lines treated with BSA control, IFNα, IFNβ, or IFNγ (all at 1000 U/ml) for 72 h. Data shown are mean ± sd (*n* = 3 biologically independent experiments) and were compared using one-way ANOVA with Dunnett correction for multiple comparisons, and adjusted *p*-values are shown. Source data are provided as a Source Data file.

contribution of these characteristics remains unclear. For instance, poor responses to immune checkpoint inhibitors are associated with limited gut microbiome diversity and gut inflammation[17–22], induced expression of immune checkpoint molecules (including PD-L1, LAG-3, TIM-3)[23–27], and immune cell exclusion within the tumor microenvironment[28]. Mechanisms driving some of these putative markers of resistance have also been reported. For instance, induction of immune inhibitory ligands such as PD-L1, LGALS9, and TNFRSF14 may reflect an adaptive negative feedback mechanism involving sustained IFNγ signaling (rather than loss of IFNγ activity)[25] while CD8 T cell exclusion from the tumor microenvironment may be caused by an immunosuppressive tumor cell secretome driven by aberrant ß-catenin, phosphatase and tensin homolog (PTEN) loss, or CDK-cell cycle signaling pathways[29–31].

In this study, we functionally dissected tumor-intrinsic mechanisms of immunotherapy resistance in a unique panel of short-term melanoma cell lines, termed PD1 PROGs, and matched tumor biopsies derived from 18 patients progressing on PD1 inhibitors, either alone or in combination with ipilimumab. We identify three distinct programs of resistance to immunotherapy in melanoma. These programs include (1) the concurrent genetic and epigenetic disruption of the MHC proteins, (2) diminished expression of wild-type antigens via de-differentiation and tumor-intrinsic IFNγ signaling, and (3) immune cell exclusion associated with PTEN loss and brain metastasis. These resistance mechanisms reveal potential salvage treatment options, including the restoration of MHC expression via epigenetic regulators, stimulation of innate immunity, and combination therapies targeting de-differentiated melanoma.

## Results

### Patient and sample characteristics

This study identified 18 patients who progressed on PD1 inhibitor monotherapy (nivolumab or pembrolizumab; $n = 9$) or combination PD1 and CTLA4 inhibitors ($n = 9$) (Supplementary Table S1); 11 patients had innate progressive disease, and 7 patients had acquired resistance (Supplementary Table S2). Of the seven with acquired resistance, five progressed after an initial partial response, and two progressed after stable disease as the best response. Ten patients received prior systemic therapy, most commonly BRAF-targeted therapy (combination dabrafenib/trametinib, $n = 8$; triple combination BRAF, MEK, and CDK4/6 inhibition, $n = 1$) and/or immunotherapy (ipilimumab $n = 1$, pembrolizumab $n = 1$, nivolumab+IDO inhibitor $n = 1$) (Supplementary Table S2). The median age of the patient cohort was 65.5 years (range 31–81); 33% (6/18) were male, 56% (10/18) had a BRAF$^{V600}$ mutation, and 33% (6/18) had an NRAS mutation (Supplementary Table S1).

A total of 21 tumor biopsies were collected from these 18 patients at the time of immune checkpoint inhibitor progression. Twenty-two short-term melanoma cell lines, termed PD1 PROGs, were derived, each from the 21 tumor biopsies, with two melanoma subclones (WMD-084#1 and WMD-084#2) generated from a single tumor biopsy (Supplementary Table S2). All progressing lesions were classified as either innate (13/22 PD1 PROGs) or acquired progressing tumors (9/22 PD1 PROGs) based on the in vivo response of each lesion to therapy (Supplementary Table S2). Innate progressing lesions were defined as pre-existing metastases that never underwent tumor shrinkage and increased in size or new metastases identified within 6 months of starting treatment. Acquired progressing lesions were defined as pre-existing tumors that initially underwent tumor shrinkage by >30% from baseline but subsequently progressed on PD1 inhibitor or new metastases identified after 6 months of starting PD1 inhibitor. It is worth noting that 10/13 innate PD1 PROGs were derived from eight patients who received prior systemic therapy (Supplementary Table S2).

### Intrinsic IFNγ signaling is more common than loss of IFNγ activity in immune checkpoint inhibitor-resistant melanoma

We examined IFNγ signaling in PD1 PROG melanoma cells treated with 1000 U/ml IFNγ for 24 h. Only one of the 22 PD1 PROG cell lines (4%) failed to respond to IFNγ based on the analysis of the Reactome_Interferon_Gamma_Signaling transcriptome gene set[32] (Fig. 1A). IFNγ signaling was disrupted in the SCC16-0016 PD1 PROG cell line because of a genomic deletion/fusion event involving the *JAK2 and INSL6* genes on chromosome band 9p24.1 (Supplementary Fig. S1A–C). This genetic deletion resulted in the loss of *JAK2* transcript and protein expression (Fig. 1B, C) and resistance to IFNγ. Thus, IFNγ treatment of SCC16-0016 cells did not induce STAT1 phosphorylation (Fig. 1C), did not promote MHC-I, MHC-II, PD-L1, or PD-L2 expression (Fig. 1D), nor induce proliferative arrest (Fig. 1E). In contrast, treatment with the type I interferons, IFNα and IFNß, induced potent cell arrest and MHC-I expression in this PD1 PROG cell line (Fig. 1F). The transient reintroduction of FLAG-tagged wild-type JAK2 into the SCC16-0016 PD1 PROG cell line restored IFNγ-mediated induction of MHC-I (Supplementary Fig. S1D).

The remaining 21 PD1 PROG melanoma cell lines responded to IFNγ exposure (Fig. 1A), and six of these cell lines (6/21; 29%; 4 innate resistant) displayed intrinsic IFNγ signaling in the absence of exogenous IFNγ. These six cell lines showed Reactome_IFNγ_Signaling ssGSEA scores above the minimum IFNγ signaling scores in IFNγ-treated, responsive PD1 PROG cell lines ($n = 21$; not including SCC16-0016) (Fig. 2A). Gene set enrichment analysis confirmed enrichment of Hallmark interferon transcriptome gene sets, in the absence of IFNγ stimulation, in these six cell lines, along with strong enrichment of mesenchymal and invasive signatures. This was accompanied by the loss of Hallmark proliferative transcriptome signatures, including the estrogen response and oxidative phosphorylation gene sets (Fig. 2B, Supplementary Data S1). In keeping with this and recent literature[33], all six PD1 PROGs with intrinsic IFNγ activity displayed features of de-differentiation, including variable accumulation of the receptor kinase AXL, downregulation of the MITF and SOX10 transcription factors, and loss of their transcription target, melanoma-specific antigen Melan-A (Fig. 2C, Supplementary Fig. S2A). The PD1 PROGs with high intrinsic IFNγ activity were also independently classified as de-differentiated (i.e., undifferentiated or neural crest-like) based on the transcriptome signatures of the four progressive melanoma differentiation states: melanocytic, transitory, neural crest-like, and undifferentiated[34] (Fig. 2C).

### Melanoma-intrinsic IFNγ signaling is immune suppressive

Gene expression analysis of the six PD1 PROGs with intrinsic IFNγ signaling versus the remaining 15 PD1 PROG melanoma cell lines (minus the JAK2-mutant SCC16-0016 PD1 PROG) confirmed higher transcript expression of IFNγ-regulated transcription factors (IRF1/4), immune inhibitory ligands *PDCD1LG2*, *LGALS9* (PD-L2 and galectin 9, respectively) and MHC-I antigen presentation effectors (*TAP1*, *TAPBP*, *HLA-B*, *HLA-C*) in the PD1 PROGs with intrinsic IFNγ activity (FDR-adjusted $p$-value < 0.05; Supplementary Data S2). We also confirmed that most of the six PD1 PROGs with elevated baseline IFNγ activity accumulated high levels of IRF1, PD-L1, PD-L2, and MHC-I (Fig. 2D, Supplementary Fig. S2B). The six PD1 PROGs with elevated IFNγ signaling also produced an inflammatory secretome in the absence of exogenous IFNγ. They secreted elevated levels (FDR-adjusted $p$-value < 0.1) of proinflammatory chemokines (CCL1 (I-309), CCL5 (RANTES), CCL7 (MCP3), CCL11 (Eotaxin), and CXCL10 (IP-10)), cytokines (IL-3, IL-6, IL-7, IL-15, TNF-α, GM-CSF, and G-CSF), and the PDGF-BB and FGF-2 growth factors (Fig. 2E, Supplementary Data S3). Most secreted factors (TNFα, GM-CSF, G-CSF, PDGF-BB, CCL5, CCL11, IL-3, IL-6) were not induced by exogenous IFNγ in our panel of melanoma cells (Fig. 2F, Supplementary Data S4), confirming the complex inflammatory signaling profile of de-differentiated PD1 PROGs with intrinsic IFNγ signaling.

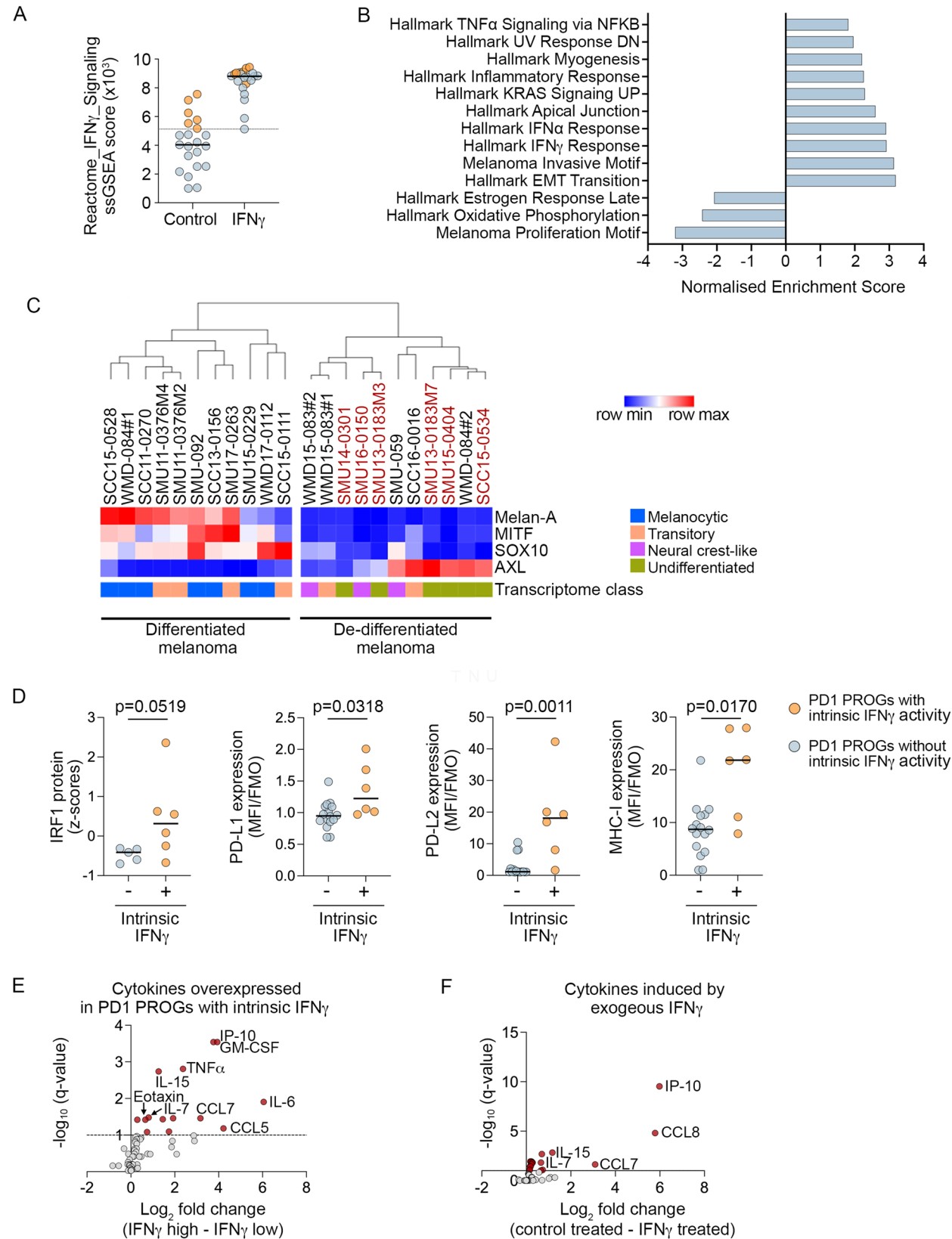

To examine the impact of intrinsic IFNγ signaling within the tumor microenvironment, we applied multiparameter flow cytometry analysis of PD1 PROG tumor dissociates. We had available 19 tumor dissociates matched with our PD1 PROG cell lines, including five showing intrinsic baseline IFNγ activity. Although the numbers are small, we noted that PD1 PROG cells with intrinsic IFNγ activity were derived from tumor biopsies with a trend toward lower CD45+ cell content (3/5 tumor dissociates with <10% CD45+ cells) and high macrophage content (2/5 dissociates with >35% macrophages) (Fig. 3A). There was also a diminished frequency of activated/exhausted CD8+ T cell subsets, including PD1++Tbet+GzmB+Ki67+, 4-1BB+, CTLA-4+LAG3+, Eomes+Tbet+ CD8+ T cells (Fig. 3B, Table 1, Supplementary Fig. S3). The diminished

**Fig. 2 | Intrinsic IFNγ signaling in seven PD1 PROG cell lines. A** ssGSEA score for Reactome_IFNγ_Signaling in BSA control and IFNγ-treated PD1 PROG cell lines with ($n = 6$; orange) and without ($n = 15$; blue, not including SCC16-0016) intrinsic IFNγ activity. The dotted line aligns with the lowest ssGSEA score in the IFNγ-treated PD1 PROGs, and solid lines indicate the median ssGSEA scores. **B** Subset of top-scoring gene sets (GSEA Pre-Ranked; Hallmark gene set collection and melanoma-specific transcriptome signatures) in PD1 PROGs with ($n = 6$) compared to without ($n = 15$) intrinsic IFNγ activity (Supplementary Data S1). **C** Hierarchical clustering of PD1 PROG cell lines ($n = 22$) with Euclidean distance based on protein expression of Melan-A, MITF, SOX10, and AXL. The transcriptome melanoma clusters defined according to ref. [34] are also shown. PD1 PROG cell lines with intrinsic IFNγ signaling ($n = 6$) are highlighted in red. **D** Plots showing IRF1 protein expression (derived from the densitometric normalized protein data after $\log_2$ transformation and $z$ score calculation) in PD1 PROGs with ($n = 6$) and without ($n = 5$) intrinsic IFNγ activity. The relative cell surface expression (median fluorescence intensity stained divided by fluorescence minus one control, MFI/FMO) of PD-L1, PD-L2, and MHC-I in PD1 PROGs with ($n = 6$) and without ($n = 16$) intrinsic IFNγ activity. Data compared using two-sided Mann–Whitney test, $p$-values shown. **E** Differentially expressed secreted cytokines (FDR-adjusted $p$-value < 0.1, dotted line) in control (BSA)-treated PD1 PROGs with (IFNγ-high; $n = 6$) or without (IFNγ-low; $n = 15$) intrinsic IFNγ signaling are highlighted in red. The comparison of secreted cytokine expression was performed using $\log_2$ transformed fluorescence intensity values (Supplementary Data S3). **F** Differentially expressed secreted cytokines (FDR-adjusted $p$-value < 0.1) in control (BSA)-treated PD1 PROGs versus IFNγ-treated PD1 PROG cell lines ($n = 21$; 1000 U IFNγ/ml for 24) are highlighted in red. The comparison of cytokine expression was performed using $\log_2$ transformed fluorescence intensity values (Supplementary Data S4). Source data are provided as a Source Data file.

frequency of activated CD8$^+$ T cell subsets did not correlate with the presence of regulatory (CD4$^+$FOXP3$^+$) T cells in PD1 PROG tumors, irrespective of the IFNγ intrinsic activity (Fig. 3C). Thus, tumor cells with intrinsic IFNγ activity occur in a tumor microenvironment with a paucity of activated CD8$^+$ effector T cells.

We compared these data to the SCC16-0016 (*JAK2*-mutant) tumor dissociate, and as expected the tumor cells in this metastatic lesion did not express MHC-I or PD-L1 (Fig. 3D). Although, we identified only one PD1 PROG with loss of IFNγ signaling, it was interesting to note that this tumor had low frequency of CD8$^+$ T cells (14.4% of CD45$^+$ cells; bottom quartile <29%), and the highest percentage of natural killer (NK) cells in our tumor dissociate panels (11% of the CD45$^+$ cells; upper quartile >5%) (Fig. 3D). This was particularly interesting as NK cells are an innate barrier against MHC-I negative tumors, but these NK cells may be suppressed by the presence of highly activated regulatory (CD4$^+$FOXP3$^+$) T cells[35] (8.1% of CD45$^+$ cells were regulatory T cells and the CD8/Treg ratio = 1.8 was in the bottom quartile; Fig. 3D). The Treg cells in the SCC16-0016 tumor dissociate expressed markers indicative of functional activation, with >80% of Treg cells expressing the activation markers CD38, ICOS, and OX40 (Fig. 3D).

### De-differentiating melanomas have a diminished melanocytic antigen repertoire and lack antigen-experienced CD8 T cells

A total of 11/22 (50%) PD1 PROGs displayed a de-differentiated phenotype that was characterized by the downregulation of MITF, SOX10, Melan-A, and elevated expression of AXL proteins (Fig. 2C, Supplementary Fig. S2A)[10]. De-differentiating PD1 PROGs included the six PD1 PROGs with intrinsic IFNγ signaling, the *JAK2*-mutant SCC16-0016, and four additional PD1 PROGs. Analysis of matched tumor dissociates confirmed that de-differentiated PD1 PROGs were derived from tumors with less proliferative (trend toward lower % Ki67$^+$ melanoma) and more de-differentiating (increased melanoma expressing the stem cell marker NGFR) melanoma cells (Fig. 4A, Supplementary Fig. S3). Importantly, these de-differentiating tumor biopsies showed fewer activated/exhausted CD8$^+$ T cells (PD1$^{++}$Tbet$^+$ CD8$^+$ T cells) indicative of the absence of antigens (Fig. 4A).

To confirm the impact of losing wild-type melanocytic antigens on immune cell recognition, we loaded the de-differentiated SMU14-0301 or WMD-084#2 PD1 PROG cell lines with the HLA-A02 binding Melan-A peptide (AAGIGILTV). In these HLA-A2:01$^+$ melanoma cell lines, Melan-A peptide presentation induced a small but significant increase in the fraction of activated (IFNγ$^+$CD107$^+$) allogeneic HLA-A02 matched or autologous CD8$^+$ T cells (Fig. 4B). In contrast, there was no change in autologous CD8$^+$ T cell activation when differentiated (Melan-A positive) HLA-A2:01$^+$ WMD-084#1 PD1 PROG cells were pulsed with the Melan-A peptide (Fig. 4B). The differential effects of Melan-A peptide pulsing on T cell activation were not due to differences in MHC-I expression in these PD1 PROG cell lines (Fig. 4C).

Although Melan-A peptide loading restored anti-tumor immunity, reinstatement of the native antigenic repertoire was more challenging as the de-differentiated phenotype was not reversible in culture. HSP90, NGFR, AXL, and HDAC inhibitors have all been shown to variably suppress the de-differentiated melanoma phenotype, but the treatment of the de-differentiated PD1 PROG cells with HSP90 inhibitors (Ganetespib at 50 nM or 250 nM, 17-AAG at 300 nM), NGFR inhibitor (1 μM Tyrphostin AG-879), histone deacetylase inhibitor (25 nM panobinostat), or AXL inhibitor (1 μM R428) did not restore expression of differentiation markers, SOX10, MITF, and Melan-A (Supplementary Fig. S4).

### Multiple independent alterations dampen antigen presentation in immune checkpoint inhibitor resistance

Six PD1 PROG cell lines (6/22; 27%) had genetic alterations affecting MHC class I and/or class II expression, and only one of these cell lines was de-differentiated (SMU-059). The SCC13-0156 and SMU-092 PD1 PROG cell lines did not accumulate MHC-I at baseline or after IFNγ treatment (Fig. 5A), and both cell lines had loss of *B2M* transcript and protein (Fig. 5B–D), a principal component of the MHC-I complex. Whole exome and transcriptome analysis revealed an exon 1 deletion in SMU-092, while SCC13-0156 showed a frameshift mutation (c45-48delTTCT, p. S16fs*27) in the *B2M* gene (Supplementary Fig. S5). Absence of MHC-I expression was confirmed in the melanoma cells from the corresponding tissue dissociates (Fig. 5E). The importance of MHC-I was validated by examining autologous CD8$^+$ T cell activation (IFNγ$^+$/CD107$^+$ CD8$^+$ T cells) in the presence or absence of MHC-I blocking antibody in co-culture experiments with two MHC-I positive melanoma cell models (Fig. 5F).

The B2M-altered SCC13-0156 and SMU-092 melanoma cell lines and matched tumor dissociates also expressed low levels of MHC-II (Fig. 6A, B). These two cell lines expressed undetectable to low levels of the critical MHC-II transcriptional regulator *CIITA* at the transcript and protein level at baseline and post IFNγ treatment (Fig. 6C, D). We also confirmed that IFNγ-induced *CIITA* transcript expression strongly correlated with MHC-II protein (Fig. 6E). Another two cell lines (SMU17-0263 and SCC11-0270) displayed low *CIITA* transcript, CIITA protein, and MHC-II expression pre and post exogenous IFNγ treatment (Fig. 6A–E). These two PD1 PROG models showed IFNγ-induced expression of MHC-I, although MHC-I membrane accumulation was below the median accumulation of other PD1 PROG cell lines (Fig. 5A).

The low expression of *CIITA* in SMU17-0263 and SCC11-0270 PD1 PROGs was not due to sequence alterations within the gene or promoter (Supplementary Data S5) but rather reflected changes in histone acetylation. Treatment of SMU17-0263 and SCC11-0270 cell lines with the HDAC inhibitor panobinostat restored IFNγ-mediated induction of CIITA (Fig. 6F) and MHC-II (Fig. 6G). Importantly, panobinostat also increased the IFNγ-mediated induction of MHC-I in these two innate resistant PD1 PROG cell models (Fig. 6H).

Complete loss of heterozygosity across both the MHC-I and MHC-II loci was present in two PD1 PROG cell lines (SMU-059 and WMD17-0112). Importantly, restoration of the missing HLA-A02:01

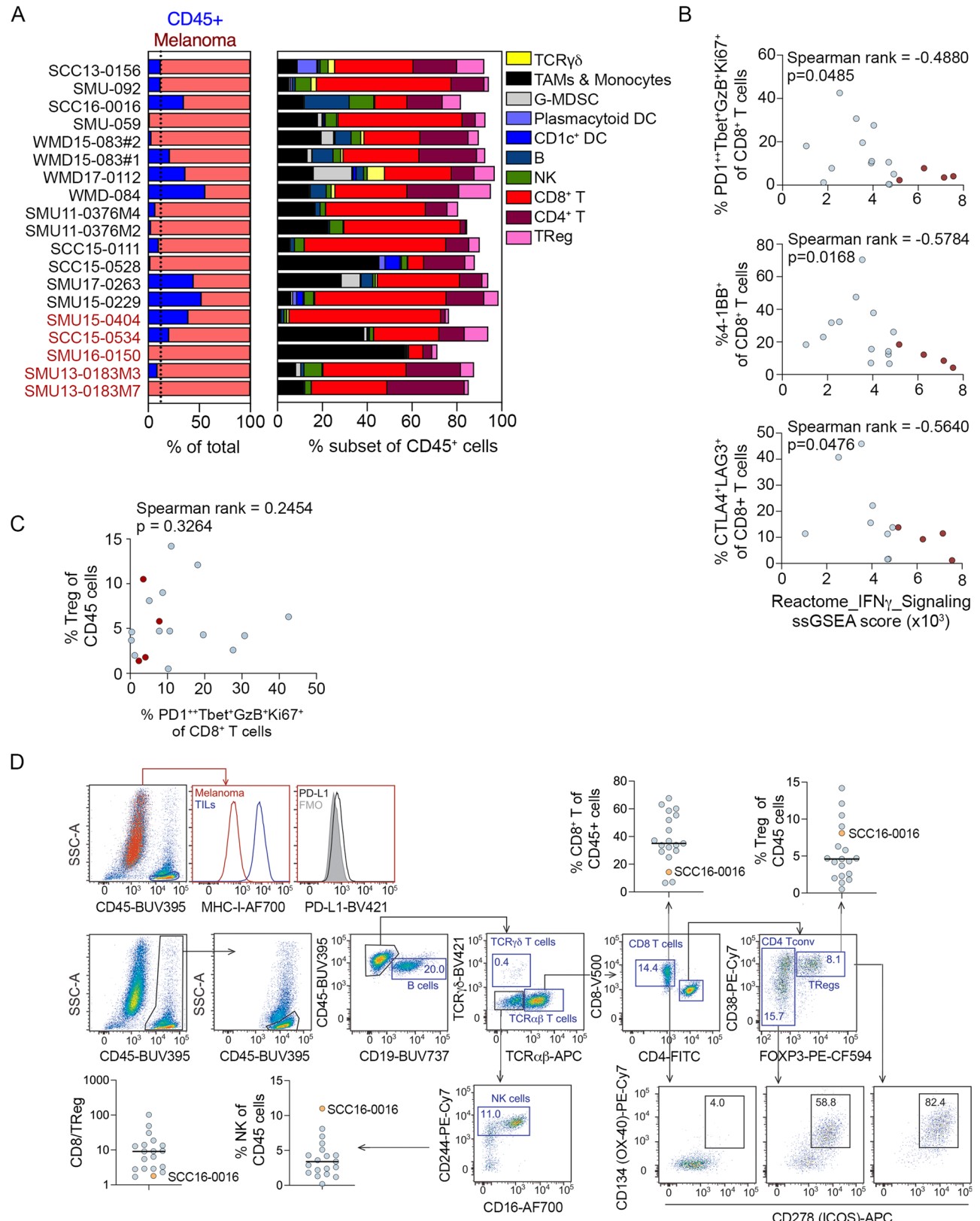

allele in the WMD17-0112 cell line reinstated detection by allogeneic matched T cells after pulsing with the HLA-A02:01 receptive Melan-A peptide (Fig. 7A). Although the magnitude of response was low, there was a significant increase in double IFNγ⁺/CD107⁺ reactive T cells when HLA-A02 was restored in the WMD17-0112 cells (mean increase 2.7×; Fig. 7A). In addition to loss of heterozygosity, 5 PD1

PROGs were derived from patients with germline HLA-A locus homozygosity, including SMU-092 (acquired resistance and B2M deficient), SMU11-0376M2 and M4 (innate resistance, and both PTEN deficient), SCC15-0528 (innate resistance, and PTEN deficient), and SMU17-0263 (innate resistance, and PTEN deficient) (Supplementary Data S6).

**Fig. 3 | Melanoma and immune cell content in PD1 PROG tumor dissociates.**
**A** Immune cell content of PD1 PROG tumor samples profiled by flow cytometry, the indicated percentage of CD45[+] cells in each dissociate (left panel) and immune cell subtypes are shown as a percentage of CD45[+] cells (right panel). PD1 PROG tumor dissociates with persistent baseline IFNγ signaling (in the matched PD1 PROG cells) are highlighted in red. The dotted vertical line indicates the median percentage of CD45[+] cells (12.2% of total cells). TCR, T cell receptor, TAMs, tumor-associated macrophages, G-MDSCs, granulocytic myeloid-derived suppressor cells, DC, dendritic cell, NK, natural killer cell, Treg, regulatory T cells. See Supplementary Table S5 for subset identification. **B** Scatterplots showing correlation of PD1[++]Tbet[+]GzB[+]Ki67[+], 4-1BB[+], and CTLA4[+]LAG3[+] CD8[+] T cell fractions in tumor biopsies with the ssGSEA scores for Reactome_IFNγ_Signaling derived from matching PD1 PROGs. Correlation was calculated using a two-sided Spearman's rank correlation coefficient, exact p-values shown. Samples highlighted in red indicate PD1 PROGs with intrinsic IFNγ signaling. **C** Scatterplots showing correlation of PD1[++]Tbet[+]GzB[+]Ki67[+] CD8[+] T cell fractions in tumor biopsies with regulatory T (Treg, FOXP3[+]) cells in the immune (CD45[+]) subset from matching PD1 PROGs. Correlation was calculated using a two-sided Spearman's rank correlation coefficient, exact p-values shown. Samples highlighted in red indicate PD1 PROGs with intrinsic IFNγ signaling. **D** Representative flow cytometric profile of SCC16-0016 tumor dissociate showing low MHC-I and PD-L1 expression in the melanoma (CD45[-]SOX10[+]) subset and presence of B (CD19[+]), NK (CD244[+]/CD16[+/-]), CD8[+], CD4[+] conventional (Tconv, FOXP3[-]) and regulatory T (Treg, FOXP3[+]) cells in the immune (CD45[+]) subset. Blue boxes show frequencies of gated cells (% CD45[+]), black boxes show the frequency of T cells expressing the activating receptors as defined by Boolean gating. Plots display flow cytometry data for the percentage of CD8[+] T, Treg, and NK cells in the CD45[+] population and the ratio of CD8/Treg in tumor dissociates (n = 19). Source data are provided as a Source Data file.

We explored the relationship between MHC-I/MHC-II expression on melanoma and immune cell content in the matched panel of tissue dissociates. The relative expression of MHC-I and MHC-II on melanoma cells within tumor dissociates was highly correlated, and all PD1 PROG cell lines with MHC-I/II alterations showed diminished MHC-I/II expression in their matched tumor dissociates (Fig. 7B). An additional three tumor dissociates displayed low MHC-I and -II expression on the melanoma cells (SCC15-0534, SCC15-0528, and WMD15-083#2), and no additional causal mechanisms were identified in the matching PD1 PROG models derived from these tumors (Fig. 7B). In melanoma with low MHC-I/II expression (defined here as ≤ median expression), there were fewer CD45[+], CD8[+], Tαβ, and macrophage cells in the tumor microenvironment (Fig. 7C). The frequency of CD4[+] and regulatory T cells did not differ according to MHC-I/II melanoma expression (Fig. 7C). In total, 12/19 tumor dissociates displayed low MHC-I and/or MHC-II expression on melanoma cells, of which eight demonstrated less than 10% immune (CD45[+]) infiltration (Fig. 7C, Table 1).

**Diminished immune cell infiltration is associated with immune checkpoint inhibitor-resistant brain tumors**
Five PD1 PROG melanoma cell lines had loss-of-function alterations affecting the *PTEN* tumor suppressor gene. The patient-matched SMU11-0376M2 and SMU11-0376M4 PD1 PROGs and SMU17-0263 (all three PD1 PROGs from brain metastases) displayed a complete loss of the *PTEN* tumor suppressor gene locus. SMU16-0150 (intrinsic IFNγ signaling, de-differentiated) showed chromosome deletions affecting *PTEN* exons 1, 2, and 3, and SCC15-0528 had a homozygous terminating Y336* mutation (c. 1008 C > G). As previously reported for PTEN-null melanomas[31], 4/5 PTEN-null melanomas showed low immune cell infiltration in the matched tumor dissociates (Fig. 3A and Supplementary Fig. S6), and all PTEN-null melanomas displayed innate resistance to immune checkpoint inhibitors (Table 1; Fisher's exact test p < 0.05), Consistent with previous reports (reviewed in ref. [36]), PTEN-null melanoma cells within tumor dissociates did not consistently display elevated PD-L1 expression, relative to TILs (Supplementary Fig. S6).

Importantly, 3/5 PTEN-null PD1 PROGs were derived from the brain, and analysis of the eight PD1 PROG cell lines derived from brain metastases revealed that all were innate progressing lesions and 5/7 matched tumor dissociates with flow cytometry data had low immune cell infiltration (CD45[+] <12% median, Table 1, Fig. 3A). Of the two remaining PD1-resistant brain melanoma tumors, high CD45[+] immune cell infiltration was associated with accumulation of immune-suppressive myeloid cells (SMU17-0263, Fig. 3A, Supplementary Fig. S7A) or elevated tumor PD-L1 expression and actively proliferating and immunosuppressive regulatory T cells, as indicated by high CTLA4, PD1, CD39, OX40, and ICOS expression[37] (SMU15-0229; Supplementary Fig. S7B, Table 1).

**Immune checkpoint inhibitor resistance mechanisms are not enriched in PRE-treatment melanoma**
We also examined whether immune checkpoint inhibitor mechanisms were enriched or selected in short-term melanoma cell models derived from patients prior to treatment with systemic therapy. We identified seven pre-treatment cell lines (PRE-melanoma), and two of these (SMU16-0570 and SCC16-0040) came from patients who subsequently went on to receive and respond to combination ipilimumab and nivolumab (Supplementary Table S3). All seven PRE-melanoma cell lines had intact IFNγ signaling, accumulated cell surface MHC-I and MHC-II molecules, and did not display intrinsic IFNγ activity (Supplementary Fig. S8). Further, only 1/7 (14%) of the PRE-treatment cell models (SMU16-0570) showed consistent features of de-differentiation that included AXL protein accumulation with the concurrent loss of MITF, SOX10, and Melan-A proteins (Supplementary Fig. S8D). Finally, the MHC-I locus was heterozygous in all seven PRE-melanoma cell lines (Supplementary Data S6), and only the SMU16-0570 PRE-melanoma, which was derived from brain metastasis, had an inactivating *PTEN* mutation (p.A192fs*20). Missense *PTEN* mutations were identified in the SCC16-0323 and WMD-031 PRE-melanoma cells (Supplementary Table S3), although these mutations are of unknown clinical significance[38].

## Discussion
A limited number of recurrent genetic effectors of immune checkpoint inhibitor resistance have so far been identified. In this study, we combined functional analysis of short-term melanoma cell models with high dimensional flow cytometric profiling of matched tumor samples from melanoma patients progressing on PD1 inhibitor monotherapy or in combination with the CTLA4 inhibitor, ipilimumab. We identified three key immune checkpoint inhibitor resistance programs in melanoma. First, disruption of MHC class I and/or II (concurrent disruption in four PD1 PROGs) was found in 6/22 (27%) PD1 PROG melanomas and was driven by independent mechanisms, including genetic alterations affecting *B2M*, epigenetic dysregulation of the MHC-II transcription regulator CIITA, and loss of heterozygosity across the chromosome 6p MHC-I/II locus. Five PD1 PROG melanoma cell lines displayed homozygosity of HLA-A alleles concurrently with other resistance effectors, including loss of PTEN (4/5) and loss of B2M (1/5). MHC class I and II homozygosity is associated with reduced survival after immune checkpoint inhibitor therapy, although this effect is influenced by the expression of the HLA allele, the HLA supertype, and the tumor mutation burden[12]. Second, suppression of wild-type antigen expression via an intractable de-differentiation program was common and identified in 11/22 (50%) PD1 PROGs. De-differentiation was driven by a sustained tumor-intrinsic IFNγ signaling program in 5/11 de-differentiated PD1 PROGs. This persistent IFNγ activity is associated with an altered melanoma secretome and favored an immune-suppressive environment enriched for exhausted CD8[+] T cells and elevated expression of immune inhibitory ligands PD-L1,

**Table 1 | Immune checkpoint inhibitor resistance landscape in melanoma**

| PD1 PROG (n = 22) | Biopsy site | Resistance mechanisms in PD1 PROG melanoma cell lines | | Tumor cells | Tumor microenvironment |
|---|---|---|---|---|---|
| | | DNA | RNA/Protein | (Dissociate) | (Dissociate) |
| SCC16-0016 | Pancreas | JAK2 loss | De-differentiation | Low MHC-I/II | Low CD8, low CD8/Treg |
| SMU16-0150 | Scalp | PTEN loss | Intrinsic IFNγ, de-differentiation | High MHC-I, high PD-L1, low MHC-II | Low CD45+, high Mφ, low CD8 T |
| SMU15-0404 | Axillary LN | | Intrinsic IFNγ, de-differentiation | High MHC-I/II | Low activated CD8 T |
| SCC15-0534 | Neck | | Intrinsic IFNγ, de-differentiation | Low MHC-I/II, High PD-L2 | High Mφ, high Tregs, low CD8/Treg |
| SMU13-0183M3 | Brain | | Intrinsic IFNγ, de-differentiation | High MHC-II | |
| SMU13-0183M7 | Brain | | Intrinsic IFNγ, de-differentiation | Low MHC-I, high MHC-II | Low CD45+ |
| SMU14-0301 | Retroperitoneal LN | | Intrinsic IFNγ, de-differentiation | NA | NA |
| WMD-084#2 | Ovaries | | De-differentiation | NA | NA |
| WMD15-083#1 | Small bowel | | De-differentiation | High PD-L2 | Low activated CD8 T |
| WMD15-083#2 | Large colon | | De-differentiation | Low MHC-I/II, high PD-L2 | Low CD45+, low CD8 T, low activated CD8 T |
| SMU-059 | Flank SC | MHC-I, MHC-II LOH | De-differentiation | Low MHC-I/II, high PD-L2 | Low CD45+ |
| WMD17-0112 | Thigh SC | MHC-I, MHC-II LOH | | Low MHC-I/II, high PD-L2 | High Tregs |
| SCC13-0156 | Retroperitoneal LN | B2M loss | CIITA silencing (MHC-II low) | Low MHC-I/II | High Tregs, low CD8/Treg, high activated Tregs |
| SMU-092 | Breast | B2M loss<br>HLA-A germline homozygous | CIITA silencing (MHC-II low) | Low MHC-I/II, high PD-L2 | Low activated CD8 T |
| SCC11-0270 | Brain | | CIITA silencing (MHC-II low) | NA | NA |
| SMU17-0263 | Brain | PTEN loss, HLA-A germline homozygous | | Low MHC-I | High Mφ, high activated Tregs |
| SMU11-0376M4 | Brain | PTEN loss, HLA-A germline homozygous | | Low MHC-II | Low CD45+ |
| SMU11-0376M2 | Brain | PTEN loss, HLA-A germline homozygous | | High MHC-I, low MHC-II, high PD-L1 | Low CD45+ |
| SCC15-0528 | Thigh SC | PTEN truncation, HLA-A germline homozygous | | Low MHC-I/II, high PD-L1 | Low CD45+, high Mφ, low CD8 T, low CD8/Treg |
| WMD-084#1 | Ovaries | | | High PD-L1 | High Tregs, low CD8/Treg |
| SCC15-0111 | Brain | | | High MHC-II | |
| SMU15-0229 | Brain | | | High MHC-I/II, high PD-L1 | High activated Tregs |

LOH loss of heterozygosity, LN lymph node, SC subcutaneous, NA data not available; differentiation status based on protein analysis of MITF, AXL, and NGFR.
Low MHC-I, MHC-I tumor/TILs <0.8 (bottom quartile); high MHC-I, MHC-I tumor/TILs >1.5 (top quartile); Low MHC-II, MHC-II tumor/TILs <0.3 (median); high MHC-II, MHC-II tumor/TILs >0.3 (median); High PD-L1, PD-L1 tumor/TILs >0.8 (top quartile); High PD-L2, PD-L2 tumor/TILs >1.8 (top quartile); Low CD8, <29% CD45+ (bottom quartile); Low CD8/Treg ratio, <2.9 (bottom quartile); Low activated CD8, PD1high Ki67+ CD8 subset <1% of CD45RO+CD8 T cells (bottom quartile); Low CD45+, <2.8% of total (bottom quartile); High macrophages (Mφ), >22% of CD45+ fraction (top quartile); High Tregs, >8.1% of CD45+ (top quartile); High activated Tregs, PD1+Ki67+subset >36% of CD45RO+Tregs (top quartile).

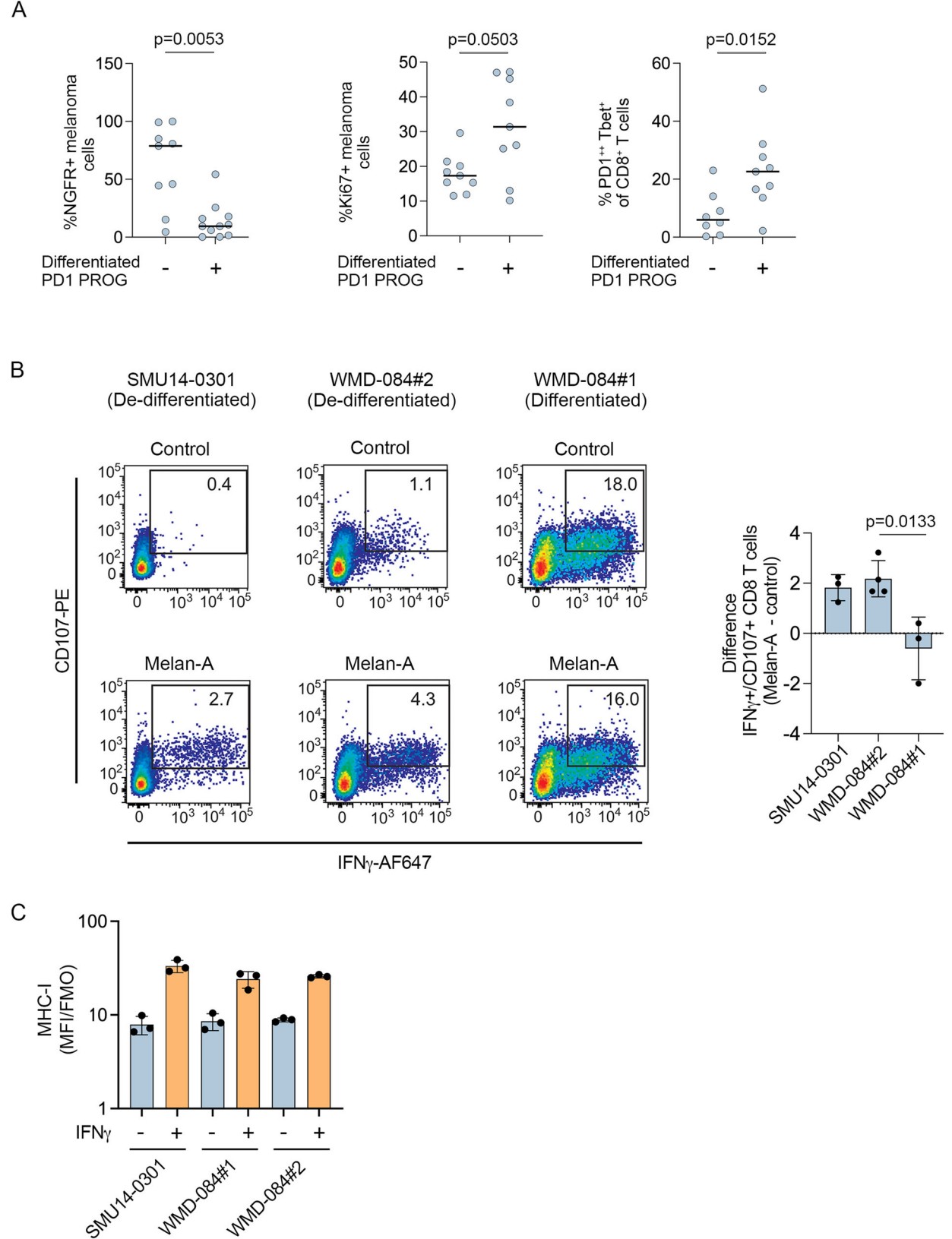

PD-L2, and galectin 9. Third, a smaller subset of PD1 inhibitor-resistant melanomas (5/22; 23%) show loss-of-function mutations in the *PTEN* gene, and this was associated with a paucity of immune cells and brain metastases. In melanoma, loss of PTEN is associated with decreased infiltration and function of tumor-infiltrating T cells, possibly due to overexpression of immunosuppressive cytokines[31], and shorter time to

brain metastasis in patients with BRAF-mutant high-risk stage III melanoma[39].

In our cohort, IFNγ signaling was disrupted in only 1/22 PD1 PROG melanoma, and this *JAK2*-mutant progressing tumor was also de-differentiated. Previous studies have also reported that alterations affecting IFNγ genes, including *JAK1/2*, *STAT1/2,* and *IFNGR1*, are rare

**Fig. 4 | Melanoma de-differentiation in immune checkpoint resistant melanoma. A** Plots showing the percentage of NGFR$^+$ melanoma cells ($n = 20$), proliferating Ki67$^+$ melanoma cells ($n = 18$) in the melanoma (CD45$^-$SOX10$^+$) subset, and PD1$^+$Tbet$^+$ CD8$^+$ T ($n = 17$) cells in the CD8$^+$ T cell subset in tumor dissociates from matched PD1 PROG cell lines. The samples are organized according to the differentiation phenotype of the matched PD1 PROG cell line. Horizontal bars indicate the median. Data compared using two-sided Mann–Whitney test, p-values shown. **B** Representative scatterplots showing reactivity of CD8$^+$ T cells (CD107$^+$/IFNγ$^+$) to the indicated PD1 PROG cell lines after pulsing with DMSO control or 10 μg/ml Melan-A peptide (AAGIGILTV) 2 h before co-culture with HLA-A02:01$^+$ CD8$^+$ T cells derived from the WMD-084 tumor dissociate. Bar graph shows the mean ± sd difference (Melan-A peptide pulsed minus control pulsed) in percentage of CD107$^+$/IFNγ$^+$ CD8$^+$ T cells ($n = 3$ for SMU14-0301 and WMD-084#1 and $n = 4$ for WMD-084#2 biologically independent experiments). The change in CD8$^+$ T cell reactivity in the patient-matched WMD-084 #1 and #2 PD1 PROG cells was compared using an unpaired, two-tailed t-test, exact p-values shown. **C** Cell surface expression (median fluorescence intensity stained divided by fluorescence minus one control, MFI/FMO) of MHC-I in the indicated cell lines at 72 h after treatment with BSA control or IFNγ (1000 U/ml). Mean ± sd average of three biological replicates shown for each cell line. Source data are provided as a Source Data file.

(~5%), and although they have been identified in post-PD1 inhibitor progression melanoma samples[9,16], these mutations have also been found in responding and non-responding melanoma patients[8,40].

Collectively, our analysis confirms that the production or presentation of antigens is disrupted in the majority (16/22; 73%; Supplementary Fig. S9) of PD1 inhibitor-resistant tumors. However, the mutually exclusive nature of the specific resistance mechanisms contributing to this deficiency in PD1 PROG cells was particularly striking (Supplementary Fig. S9). Specifically, in the 19 PD1 PROGs with an identified resistance effector, MHC, de-differentiation and PTEN loss show a tendency toward mutual exclusivity in our panel of PD1 PROG melanomas (odds ratio for the co-occurrence of de-differentiation and MHC loss, 0.06; odds ratio for the co-occurrence of de-differentiation and PTEN loss, 0.1; and odds ratio for the co-occurrence of MHC and PTEN loss, 0.5; Table 1).

Our work indicates that loss of both MHC-I and MHC-II pathways often co-occurs in immune checkpoint inhibitor resistance, although we hypothesize that the relative contribution of each MHC molecule will reflect the profile of antigens expressed by the tumor cells. For instance, tumors expressing antigens recognized by CD4$^+$ T cells may require MHC-II loss for PD1 inhibitor escape[41], in line with a direct CD4$^+$ T cell cytotoxicity against cancer cells[42]. Tumor-specific MHC-I and MHC-II expression is correlated with T cell infiltration and response to immune checkpoint inhibitors in many different tumor types[43–46]. While the critical role of MHC-I loss in immune evasion is well documented[47], the impact of MHC-II loss is less defined. The emerging consensus is that MHC-II directly presents tumor antigens to activate CD4$^+$ T cells within the tumor microenvironment, and the requirement for MHC-II-mediated CD4$^+$ T cell activation in PD1 inhibitor responses has been confirmed in non-small cell lung carcinomas and classical Hodgkin's lymphoma[45,48].

De-differentiation of melanoma is also an important modulator of targeted therapy and immunotherapy response. This program can be driven by cytokines including TNFα, TGFß, and IFNγ and may be reversible upon removal of the stimuli[13,49]; thus, the transient induction of de-differentiation is reflective of an active immune response[33]. There also exists a stable and potentially irreversible melanoma de-differentiation state, as is the case for the de-differentiated PD1 PROG melanomas in this study, and this stable program is associated with resistance to targeted and immune checkpoint inhibitor therapies[50,51]. Thus, it is likely that the nature and duration of the de-differentiation switch, and the contribution of melanocytic antigens to immune-mediated tumor response[13], determine the role of this program in treatment resistance. We have also previously reported that de-differentiation is associated with the suppression of MHC-I expression[10].

The functional and multidimensional characterization of immune checkpoint inhibitor resistance in this study has additionally revealed potential combination and salvage therapies to restore treatment sensitivity. For instance, disruption of MHC-I and/or MHC-II antigen presentation due to genetically based resistance mechanisms such as *JAK1*, *JAK2*, and *B2M* loss-of-function mutations or epigenetic down-regulation of *CIITA* can be circumvented by activating innate immune receptors using Toll-like receptor agonists or dsRNA analogues[52–54]. We also show that the HDAC inhibitor panobinostat restored IFNγ-mediated induction of MHC-I and MHC-II and may potentiate anti-PD1 responses by upregulating antigen presentation molecules under circumstances of epigenetic silencing. Certainly, this combination improved response and prolonged the survival of glioma and B-cell lymphoma mouse models by inducing MHC expression[55,56]. Several other strategies, including activating the innate immune system with CD40 agonists[57] and the use of chimeric antigen receptor (CAR)-based adoptive T cell therapy[58], have shown promising results in patients with impaired MHC class I or II antigen presentation. Conversely, restoring immune checkpoint inhibitor responses in de-differentiated tumors will be more challenging, and combination treatments aimed at interfering with the initial microenvironment-driven phenotype switch or preventing the stabilization of the de-differentiated state are worth exploring. De-differentiated melanomas also show increased sensitivity to iron-dependent oxidative stress[34], and combining immunotherapies with ferroptosis-inducing agents may improve immune checkpoint inhibitor efficacy. These additional treatment strategies identified by this analysis may assist in improving the outcomes of PD1-resistant melanoma patients.

## Methods

### Patients and treatment

Eighteen patients with stage IV melanoma with progressive disease while on treatment with immunotherapy at Melanoma Institute Australia (MIA) and Westmead Hospital between November 2013 and July 2018 were included in this study. Written consent was obtained from all patients and research complied with ethical regulation (Human Research ethics approval from Royal Prince Alfred Hospital—Protocol X15-0454 and HREC/11/RPAH/444). Patients were treated with either pembrolizumab 2 mg/kg every 3 weeks, nivolumab 3 mg/kg every 2 weeks, nivolumab 1 mg/kg plus ipilimumab 3 mg/kg every 3 weeks for four doses, followed by nivolumab 3 mg/kg every 2 weeks or pembrolizumab 2 mg/kg in combination with ipilimumab 1 mg/kg every 3 weeks followed by pembrolizumab 2 mg/kg every 3 weeks. Patient demographics and clinicopathologic features, including Eastern Cooperative Oncology Group (ECOG) performance status, lactate dehydrogenase (LDH) levels at baseline, BRAF and NRAS mutation status, American Joint Committee on Cancer (AJCC) stage Eighth Edition[59], and disease distribution were collected. Investigator-determined objective response was assessed radiologically with computed tomography (CT) scans alone or where indicated, with magnetic resonance imaging (MRI) of the brain, at two to three monthly intervals using irRECIST[60].

### Tissue processing and cell isolation

Fresh tumor biopsies were collected from melanoma patients following surgical resection. Tumor biopsies were enzymatically processed and dissociated into single-cell suspensions using the Tumor Dissociation Kit and gentleMACS Dissociator (Miltenyi Biotec), according to the manufacturer's instructions. Single-cell suspensions were frozen as tumor dissociates in 10% DMSO in human serum from male AB plasma (Sigma, St. Louis, MO, USA) and were plated into 24-well plates ($1 \times 10^6$ cells/well) to isolate short-term melanoma (termed PD1 PROGs)

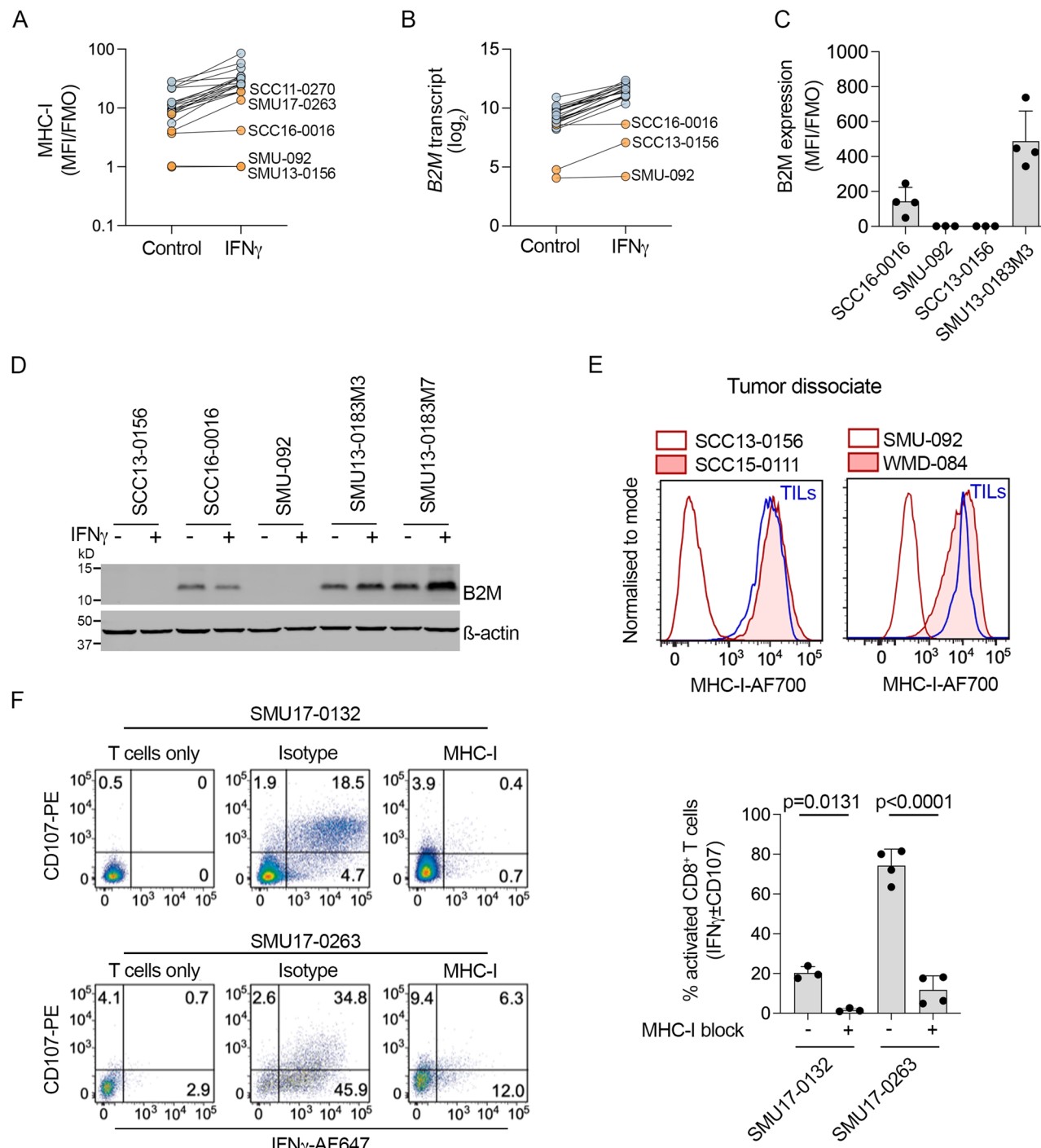

**Fig. 5 | MHC-I expression in PD1 PROG cell lines. A** Cell surface expression (median fluorescence intensity stained divided by fluorescence minus one control, MFI/FMO) of MHC-I in PD1 PROG cell lines ($n = 22$) at 72 h after treatment with BSA control or IFN$\gamma$ (1000 U/ml). Average of at least three biological replicates shown for each cell line. **B** *B2M* transcript expression at 24 h after treatment of PD1 PROG cell lines ($n = 22$) with BSA control or IFN$\gamma$ (1000 U/ml). **C** Cell surface expression (median fluorescence intensity stained divided by fluorescence minus one control, MFI/FMO) of B2M on the SCC16-0016 ($n = 4$ biological replicates), SMU-092 ($n = 3$ biological replicates), SCC13-0156 ($n = 3$ biological replicates), and SMU13-0183M3 ($n = 4$ biological replicates) PD1 PROG cell lines at baseline. Individual values and mean ± sd of biological replicates are shown. **D** Accumulation of B2M protein in 5 PD1 PROG cell lines 24 h after treatment with BSA control (−) or IFN$\gamma$ (1000 U/ml). Experiment repeated independently three times. **E** Representative histograms

showing melanoma MHC-I expression in tumor dissociates of B2M-null SCC13-0156 and SMU-092 (unshaded red histograms) compared to autologous TILs (blue). MHC-I expression on B2M-wild-type SCC15-0111 and WMD-084 is shown for comparison (shaded red histograms). **F** Representative scatterplots showing reactivity of autologous CD8$^+$ T cells (CD107$^+$/IFN$\gamma^+$) after pre-treating the SMU17-0132 (unrelated melanoma cell line) and SMU17-0263 (PD1 PROG) melanoma cells for 1 h with IgG2a isotype control or 10 µg/ml MHC-I blocking antibody. The expression of CD107 and IFN$\gamma$ in T cell mono-cultures (left panels) was used to establish the gating strategy for these experiments. Bar graph shows percentage of CD107$^+$ ± IFN$\gamma^+$ CD8$^+$ T cells (mean ± sd, (SMU-0132, $n = 3$; SMU17-0263, $n = 4$ biological replicates)) after treatment with isotype control (−) or MHC-I blocking antibody (+). The percentage CD107$^+$ ± IFN$\gamma^+$ CD8$^+$ T cells was compared using a paired, two-tailed *t*-test, exact *p*-values shown. Source data are provided as a Source Data file.

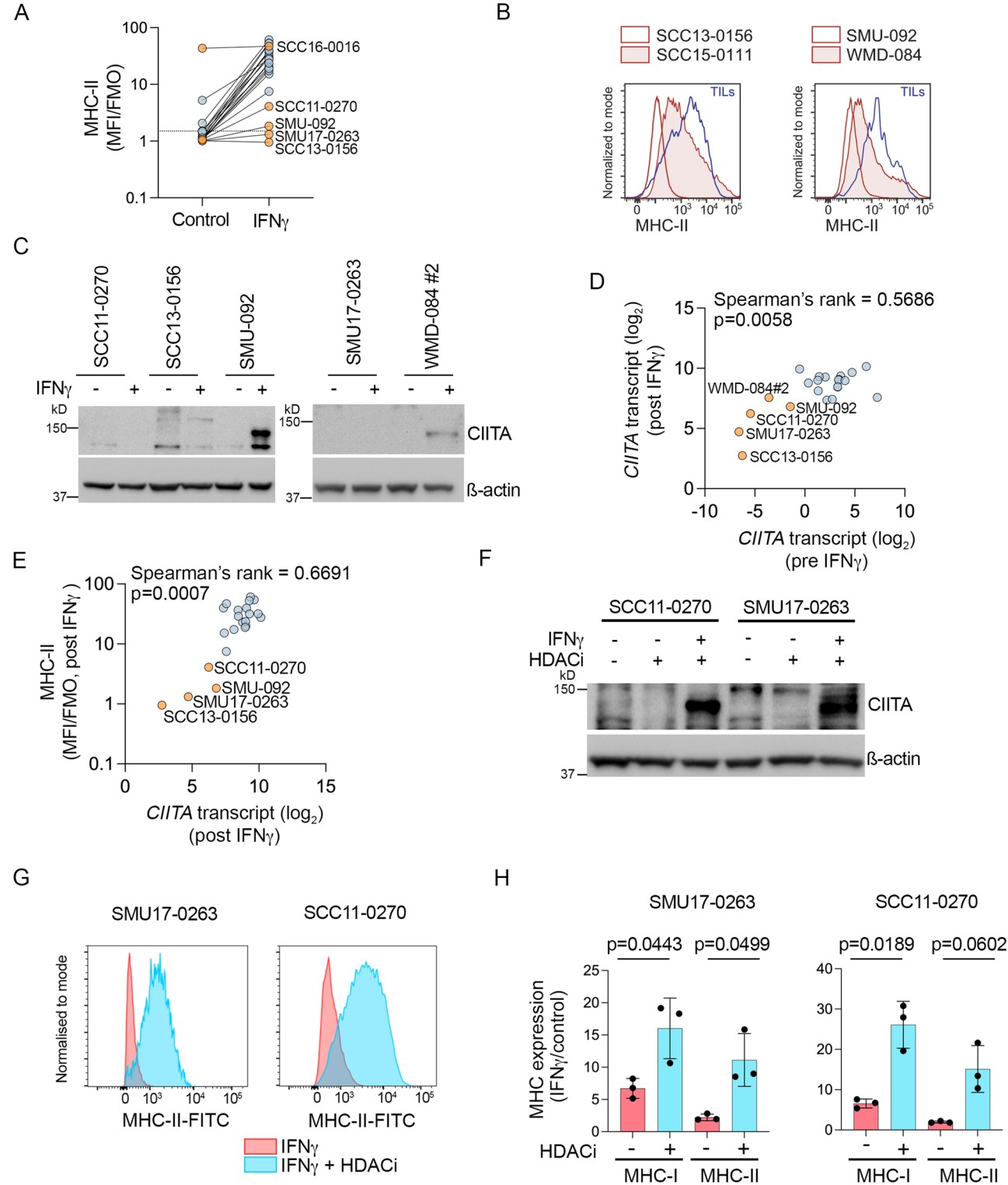

and tumor-infiltrating lymphocyte (TILs) cultures. The WMD-084#1 and WMD-084#2 melanoma subclones were derived from separate wells seeded from a single tumor dissociate.

## Cell culture

PD1 PROG melanoma cultures were maintained in Dulbecco's Modified Eagle medium supplemented with 10% heat-inactivated fetal bovine serum (Sigma), 4 mM glutamine (Sigma), and 20 mM HEPES (Sigma) at 37 °C in 5% $CO_2$. Cell authentication and profiling of newly derived cell lines were confirmed using the StemElite ID system from Promega. All

cells tested negative for mycoplasma (MycoAlert Mycoplasma Detection Kit, Lonza, Basel).

Tumor-infiltrating lymphocytes (TILs) were cultured in TIL medium (Roswell Park Memorial Institute-1640 medium supplemented with 10% heat-inactivated human serum from male AB plasma (Sigma), 25 mM HEPES, 100 U/ml penicillin, 100 μg/ml streptomycin, 10 μg/ml gentamycin, 2 mM L-glutamine and 1000 U/mL IL-2 (PeproTech, Rocky Hill, NJ, USA), and expanded with the addition of DynaBeads Human T activator CD3/CD28 (Thermo Fisher Scientific).

**Fig. 6 | MHC-II expression is silenced in PD1 PROG cell lines. A** Expression (median fluorescence intensity of at least three biological replicates for each cell line divided by fluorescence minus one control, MFI/FMO) of MHC-II on PD1 PROGs ($n$ = 22), 72 h after treatment with BSA control or IFNγ (1000 U/ml). SCC16-0016 and PD1 PROGs with low MHC-II expression are highlighted in orange. **B** Representative histograms showing melanoma MHC-II expression in tumor dissociates of B2M-mutant SCC13-0156 and SMU-092 (unshaded red histograms) compared to autologous TILs (blue). MHC-II expression on B2M-wild-type SCC15-0111 and WMD-084 shown for comparison (shaded red histograms). **C** Accumulation of CIITA protein in 5 PD1 PROGs, 24 h after treatment with BSA control (−) or IFNγ (1000 U/ml). Experiment repeated independently at least three times. **D** Scatterplot showing correlation of *CIITA* transcript expression pre and post IFNγ treatment in 22 PD1 PROG cell lines. PD1 PROGs with low *CIITA* expression ($n$ = 5) are highlighted in orange. Correlation calculated using two-sided Spearman's rank correlation coefficient, exact $p$-values shown. **E** Scatterplot showing correlation of MHC-II expression (MFI/FMO, MFI post IFNγ treatment) with expression of *CIITA* transcript (post IFNγ treatment) in PD1 PROGs ($n$ = 22). PD1 PROGs with low MHC-II expression are highlighted in orange. Correlation calculated using two-sided Spearman's rank correlation coefficient, exact $p$-values shown. **F** Accumulation of CIITA protein in the SCC11-0270 and SMU17-0263 PD1 PROGs, 72 h after treatment with BSA control, panobinostat (HDACi, 25 nM) in the presence or absence of IFNγ (1000 U/ml). Experiment repeated independently at least three times. **G** Representative histograms showing melanoma MHC-II expression in MHC-II[low] SCC17-0263 and SCC11-0270 PD1 PROGs treated with 1000 U/ml IFNγ (shaded red histograms) or IFNγ with panobinostat (HDACi; shaded blue histograms). **H** Bar graphs show relative MHC-I and MHC-II expression (IFNγ/control-treated) in SCC17-0263 and SCC11-0270 PD1 PROGs treated with 1000 U/ml IFNγ or IFNγ with panobinostat (HDACi). Individual values and mean ± sd of three biological replicates are shown and paired, two-tailed $t$-test was used to compare the data, exact $p$-values shown. Source data are provided as a Source Data file.

For IFNγ treatment, melanoma cells were plated, and after overnight incubation, the medium was replenished, and cells were treated for 24 h with 1000 U/ml IFNγ (PeproTech) or vehicle control (0.1% bovine serum albumin (Sigma) in PBS (Gibco). The concentration of IFNγ was based on ref. [61] and induced maximal levels of MHC-I and MHC-II in titration experiments. Culture supernatants were collected for cytokine analysis, and cells were collected for RNA or DNA sequencing, immunoblotting, or flow cytometry.

Melanoma cells were treated with Ganetespib (Selleckchem; Cat No: S1159), 17-AAG (Selleckchem; Cat No: S1141), Tyrphostin AG-879 (Selleckchem; Cat No: S2816), Panobinostat (Selleckchem; Cat No: S1030), or R428 (ApexBio; Cat No: A8329) for 72 h before collecting cells for immunoblotting.

### RNA isolation, sequencing, and data processing

Total RNA was isolated from melanoma cells using the RNeasy Mini Kit (Qiagen, Hilden, Germany). cDNA synthesis and library construction were performed using the TruSeq RNA Library Prep Kit (Illumina) with paired-end 150 bp sequencing, each sample yielding 40–50 million reads. Sequencing was performed on the Illumina NovaSeqS4 platform at the Australian Genome Research Facility (AGRF) in Melbourne.

RNA sequencing data were processed using Cutadapt (version 1.9), STAR (version 2.5.2), mapped to reference genome hg19 or hg38 using GenomicAlignment (version 1.12.2), normalized using the trimmed mean of M-values (TMM) and transformed with voom to log2-counts per million as previously described in ref. [10]. Differentially expressed genes between groups were determined using TMM-voom normalized $\log_2$ counts[62] and the moderated $t$-tests (limma package version 3.52.4 in R/Bioconductor)[63]. Enrichment scores were calculated from the ranked lists using gene set enrichment analysis in pre-ranked mode (GSEA pre-ranked version 7.4.0)[64] provided by GenePattern (https://cloud.genepattern.org/)[65].

To obtain abundance values corrected for transcript lengths as required by the single-sample gene set enrichment analysis (ssGSEA[66]), RSEM was used to derive the FPKM estimates using GENCODE Genes version 26 as the reference transcript database. ssGSEA was used to derive the absolute enrichment scores using the genesets from the Molecular Signature Database version 6.2[67]. The same FPKM values were used to determine the melanoma differentiation transcriptome subtypes (undifferentiated, neural crest-like, transitory, and melanocytic) using the support vector machine "top-scoring pairs" scripts kindly provided by Dr. T. Graeber[34]. These transcriptome subtypes correspond to the melanocytic (transitory and melanocytic) and de-differentiated melanoma phenotypes (undifferentiated, neural crest-like)[68,69].

### DNA extraction and whole exome and genome sequencing

DNA was extracted from melanoma cells using the G-spin™ Total DNA Extraction Kit (Intron Biotechnology, Seongnam, South Korea) or the QIAamp DNA Mini Kit (Qiagen) and quantified using the SmartSpec Plus Spectrophotometer (Bio-Rad, CA, USA). Whole exome and genome sequencing of melanoma cell lines was performed on HiSeq400 or NovaSeq instrument by Macrogen or AGRF. Sequence data were processed using Cutadapt v1.9 and aligned to the GRCh37 assembly. To generate a list of high-quality variants, low-coverage variants (variant call quality <30, read depth <10; genotype quality <30) and variants in the top 5% of most exonically variable 100 base window in health public genomes and in the top 5% most exonically variable genes regions (1000 Genomes Project), and common polymorphisms (≥5% frequency), were removed using QIAGEN Digital Insights (https://variants.ingenuity.com/qci/; Qiagen, Venlo, Netherlands). The likelihood that immunotherapy resistance programs were mutually exclusive or co-occurrent was determined as described previously[70].

### Flow cytometry

Cryopreserved tumor dissociates were available for 19 PD1 PROG samples. These dissociates were thawed into TIL medium containing 20 µg/ml DNAse type II-S (Sigma) and washed prior to staining. Staining of all samples was performed in flow cytometry buffer (PBS supplemented with 5% FBS, 10 mM EDTA, and 0.05% sodium azide). Cells were incubated for 30 min on ice with fluorescently labeled monoclonal antibodies (Supplementary Table S4) and Fc block (BD Biosciences) to prevent non-specific staining[46]. For intracellular staining, cells were fixed and permeabilized using the Transcription buffer Fixation/Permeabilization kit (Thermo Fisher Scientific), stained with antibodies against intracellular proteins and Fc block in permeabilization buffer, and washed extensively. Prior to the acquisition, cell viability was determined by staining cells with either 5 µM DAPI or Live Dead near-IR fixable dye (both from Thermo Fisher Scientific).

Samples were acquired on a 5-laser BD LSRFortessa X20 flow cytometer (BD Biosciences). FlowJo v10 software (BD) was used for data analysis. At least 10,000 live events were acquired for cell lines, while all available events were acquired for tumor dissociate analysis. Relative marker expression levels were calculated by dividing the median fluorescence intensity (MFI) of the antibody-stained sample by the unstained control. For dissociated tumors stained with multiple antibodies, marker expression levels were determined by dividing the MFI in the respective channel by the fluorescence minus one control (FMO, full staining omitting the antibody of interest). In some cases, melanoma marker expression was calculated as a score relative to tumor-infiltrating lymphocytes (MFI melanoma/TILs), as detailed in the figure legends. For details on the gating strategy and analysis of dissociated tumors, including identification of melanoma cells and immune subsets and expression of antigen-presenting molecules and immune checkpoints[46]; see Supplementary Fig. S10 and Supplementary Table S5.

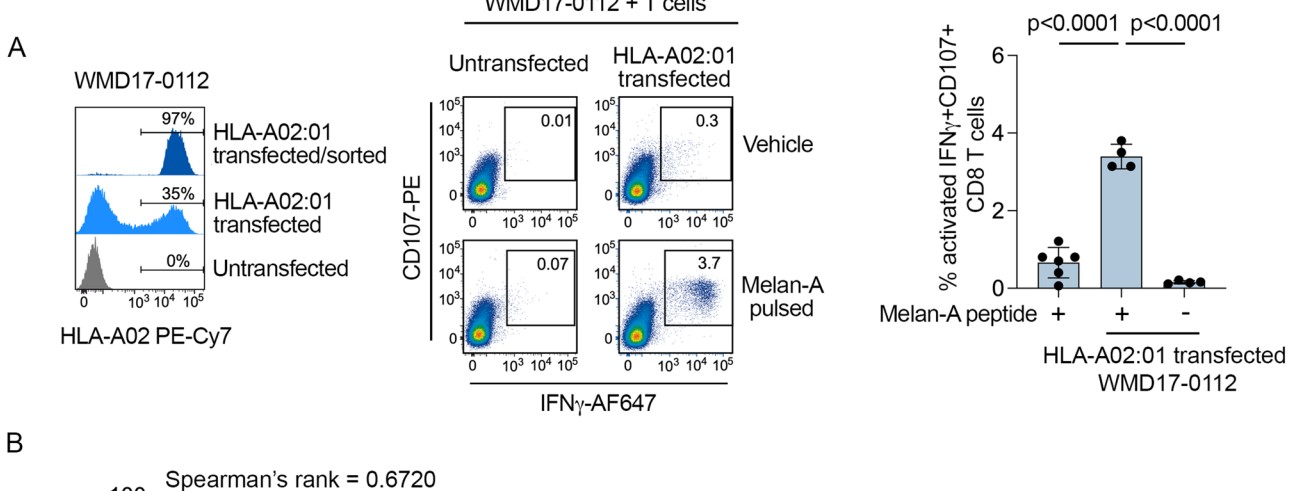

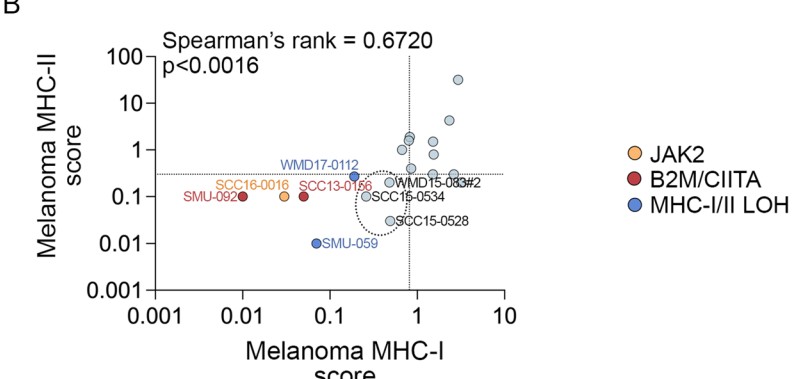

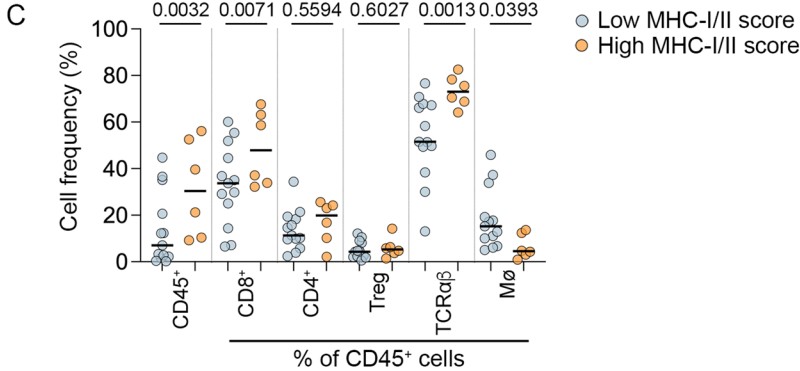

**Fig. 7 | Loss of MHC antigen presentation molecules. A** Representative histograms showing expression of HLA-A02:01 in the WMD17-0012 PD1 PROG melanoma cells (HLA-A02:01 LOH, Melan-A$^{null}$). Untransfected cells are compared to cells transfected with HLA-A02:01 pre and post flow sorting (left panel). Reactivity of allogenic HLA-A02:01 matched CD8$^+$ T cells (CD107$^+$/IFNγ$^+$) to untransfected or HLA-A02:01-transfected WMD17-0112 cells pulsed with 10 μg/ml Melan-A peptide (AAGIGILTV) or vehicle control, 1.5 h before co-culture (middle panel). Bar graph shows percentage (individual values and mean ± sd) of reactive CD107$^+$ IFNγ$^+$ CD8$^+$ T cells co-cultured with untransfected/Melan-A pulsed (+) WMD-0112 cells (n = 6 biological replicates) and HLA-A02:01-transfected Melan-A pulsed (+) or HLA-A02:01 vehicle treated (−) WMD17-0112 cells (n = 4 biological experiments). Data compared using one-way ANOVA with Tukey's multiple comparison test, adjusted p-values are shown. **B** Scatterplot showing correlation of MHC-II and MHC-I

expression (expression score relative to TILs) in PD1 PROG tumor dissociates (n = 19). Correlation calculated using two-sided Spearman's rank correlation coefficient, p-value is shown. Tumor samples with established alterations in JAK2 (SCC16-0016), MHC-I/II (WMD17-0112, SMU-059), B2M/CIITA (SMU-092, SCC13-0156) are highlighted and color-coded. Three tumors with low MHC-I/-II expression on melanoma without causal mechanisms are circled. Dotted lines indicate median MHC-I and MHC-II scores. **C** Percentage of CD45$^+$ cells and frequency of CD8$^+$, CD4$^+$ and regulatory T cells (Treg), macrophages (Mø), and TCRαβ cells (as a percentage of CD45$^+$ cells) in PD1 PROG tumor dissociates with high MHC-I and/or MHC-II score (above median; n = 6) vs low MHC-I and/or MHC-II score (below median; n = 13). Data compared using one-way ANOVA corrected for multiple comparisons by controlling the False Discovery Rate (q < 0.1); adjusted p-values are shown.

## Co-culture melanoma:TIL assays

Melanoma cells (1 × 10$^5$) were cultured 1:1 with autologous or allogeneic TILs in a 24-well plate in 0.5 ml of TIL medium. One and a half hours post co-culture, 5 μg/ml of Brefeldin A, 5 μg/ml monensin (both from Sigma), and anti-CD107a (Supplementary Table S4)

were added and co-cultures were incubated overnight. After staining for cell surface markers, samples were fixed, permeabilized (using Fixation/Permeabilization kit, Thermo Fisher), and stained for intracellular cytokines in the presence of Fc block (BD Biosciences).

For MHC-I blocking, melanoma cells were pre-treated with 10 μg/ml anti-HLA-ABC or IgG2a isotype control antibodies (Supplementary Table S4) 1 h prior to co-culturing melanoma cells with autologous TILs.

For Melan-A peptide loading, $1 \times 10^5$ melanoma cells were pulsed with 10 μg/ml Melan-A peptide (AAGIGILTV, Auspep, Australia) or DMSO for 1.5 h before co-culture with TILs. Determination of a positive anti-tumor response in co-culture experiments was based on criteria defined in ref. [71], and included a difference of >0.1% for double positive CD107/IFNγ T cells from the background (i.e., co-culture experiments with untransfected or unpulsed cells).

## HLA typing
MHC typing for Class I (HLA-A, -B, -C) and Class II (DRB1, 3, 4, 5, DQA1, B1, DPB1) was performed by an American Society for Histocompatibility and Immunogenetics-accredited laboratory at the Institute for Immunology and Infectious Diseases, Murdoch University (Western Australia), using locus-specific PCR amplification of genomic DNA as previously described in ref. [72].

## Immunoblotting
Total cellular proteins were extracted, resolved on 8–12% SDS-polyacrylamide gels and transferred to Immobilon-FL membranes (Millipore) as previously described[10]. Western blots were probed with antibodies against ß2-microglobulin (1:1000; D8P1H; Cell Signaling Technology; Cat No. 12851), CIITA (1:1000; Cell Signaling Technology; Cat No. 3793), STAT1 (1:1000; 9H2; Cell Signaling Technology; Cat No. 9176), phospho-STAT1 Ser727 (1:1000; D3B7; Cell Signaling Technology; Cat No. 8826), JAK1 (1:1000, Cell Signaling Technology; Cat No. 3332), JAK2 (1:1000, E4Y4D; Cell Signaling Technology; Cat No. 74987), IRF1 (1:1000; D5E4; Cell Signaling Technology; Cat No. 8478), AXL (1:200; R&D systems; Cat No. AF154), MLANA/Mart-1 (1:1000; Cell Signaling Technology; Cat No. 34511), MITF (1:1000; C5; Calbiochem; Cat No. OP126L), SOX10 (1:1000; D5V9L; Cell Signaling Technology; Cat No. 89356), and ß-Actin (1:6000; AC-74; Sigma-Aldrich; Cat No. A5316). Secondaries used were IRDye 680RD Goat anti-Mouse IgG, IRDye 680RD Donkey anti-Goat IgG, and IRDye 800CW Goat anti-Rabbit IgG (1:20,000; LI-COR, Lincoln, NE) for fluorescent detection and Goat anti-Mouse Immunoglobulin HRP and Goat anti-Rabbit Immunoglobulin HRP (1:6000; Agilent Dako, Santa Clara, CA) for chemiluminescence detection. Where indicated, membranes were incubated with REVERT 700 total protein stain (LI-COR, Lincoln, NE). Blots were imaged using the Odyssey CLx imaging system or ChemiDoc MP. Membranes were washed with Tris buffered saline with tween and blocked using LI-COR Odyssey blocking buffer or 5% skim milk. Protein expression data were normalized to ß-actin or to the REVERT 700 total protein stain.

## Cytokine analysis
Undiluted culture supernatants (collected 24 h after treatment with 1000 U/ml IFNγ or vehicle control) were profiled using the 65-plex Human Cytokine/Chemokine Discovery Assay (HD64, Eve Technologies) as previously described[73]. Fluorescence intensity values which reflect the amount of proteins in the samples were $\log_2$ transformed and differential cytokine expression was carried out using moderated *t*-tests (limma package version 3.52.4 in R/Bioconductor)[63].

## Cell proliferation assay (IncuCyte)
Melanoma cells were seeded into 96-well flat bottom clear plates ($2 \times 10^3$ cells/well), and after overnight incubation, the medium was replenished, and cells treated with 1000 U/ml IFNα, IFNβ, or IFNγ (PeproTech) or vehicle control in triplicates. Cells were analyzed on the IncuCyte ZOOM live cell imaging system. Six images per well were taken every 4 h using the default software parameters for a 96-well plate (Corning) with a 10× objective. The IncuCyte software was used to calculate mean confluence from four non-overlapping bright phase images for each well and the mean of the biological triplicates.

## Expression constructs
HLA-A02 was cloned from a human patient melanoma cell line. cDNA was generated using SuperScript™ III First-Strand Synthesis System (Thermo Fisher Scientific). HLA_EX1-A-6 forward primer 5-CTCTCTAGACAGACGCCGAGGATGGCC-3 and HLA_A-ex8-1 reverse primer 5- GCTGGATCCCACACAAGGCAGCTGTCTCAC-3 were used to amplify HLA-A02. Amplicon was purified using the QIAquick PCR purification kit (Qiagen) and cloned into pCDH-CMV-MCS-EF1α-Puro expression plasmid (System Biosciences, California, USA) using BamHI and XbaI restriction sites. Constructs were screened by sequencing using pCDH screen reverse primer 5-CTGTGGGCGATGTGCGC-3 and pCDH forward primer 5-TTTAGTGAACCGTCAGATCG-3. Following sequence confirmation, melanoma cells were transfected using the Neon™ Transfections System (MPK5000; Invitrogen, Waltham, MA, USA) and the Neon™ 10 μl Kit (MPK1025; Invitrogen). Briefly, melanoma cells ($2 \times 10^5$) were mixed with plasmid DNA and electroporated using cell type optimized conditions. Cells were incubated in pre-warmed complete media and sorted for HLA-A2:01-positive cells, 24 h post transfection using the MAQSQuant Tyto.

FLAG-tagged wild-type JAK2 cloned into pcDNA3.1+ (Genscript Biotech Corp, NJ) was introduced into SCC16-0016 cells using Lipofectamine 2000 (Invitrogen). Approximately 24 h post transfection, cells were treated for 48 h with 1000 U/ml IFNγ (PeproTech) or vehicle control (0.1% bovine serum albumin (Sigma) in PBS (Gibco), then collected for flow cytometry analysis.

## JAK2 and CIITA polymerase chain reaction
PD1 PROG genomic DNA (100 ng) was used in PCR reactions. JAK2 PCR products were amplified using Taq DNA polymerase (Fisher Biotec, Wembley, WA, Australia), the JAK2_Fwd (5′-GGAATCATATACTTACCTGAG), and INSLR6_Rev (5′-GACAGTCTGGGATGTTGGAC-3′) primers and the following PCR conditions: 35 cycles of 95 °C for 30 sec, 43 °C for 30 sec, and 72 °C for 1min. CIITA primers and PCR conditions were as follows: CIITA_promoter_Fwd (5′-ATAGCTAGC GGTTGGACTGAGTTGGAGAGAAA-3′) and CIITA_promoter_Rev (5′- TTCCTCGAG CGCTGTTCCCCGGGCTCCCGCGCG CGCT-3′). CIITA promoter DNA was amplified using AccuPrime Pfx Supermix (Thermo Fisher Scientific) for 35 cycles, including 95 °C for 15 sec, 61 °C for 30 sec, and 68 °C for 30 sec. PCR products were purified using the QIAquick PCR purification kit (Qiagen) and sequenced at the AGRF with the PCR primers.

## Statistics and reproducibility
No statistical method was used to predetermine sample size; sample sizes were selected to ensure robust statistical analysis within the confines and technical parameters of each experiment. No data were excluded from analyses. The experiments were not randomized, but melanoma patients included in this study were previously randomized as participants in clinical trials. The investigators were blinded to the response annotations until they became available, and flow cytometry gating was performed prior to the analysis of results.

For statistical analysis, we used GraphPad Prism software v.9. Figure legends specify the statistical analysis used and define error bars.

## Study approval
Written consent was obtained from all patients prior to participation (Human Research ethics approval from Royal Prince Alfred Hospital—Protocol X15-0454 & HREC/11/RPAH/444).

## Reporting summary
Further information on research design is available in the Nature Portfolio Reporting Summary linked to this article.

## Data availability

The RNAseq data generated in this study have been deposited in the Sequence Read Archive under accession code PRJNA818797) and are publicly available. RNA sequencing data were mapped to reference genome hg19 (GRCh37, https://www.ncbi.nlm.nih.gov/assembly/GCF_000001405.13/) or hg38 (GRCh38, https://www.ncbi.nlm.nih.gov/assembly/GCF_000001405.26/). All other data is available within the Article, Supplementary Information, and Source Data file. Source data are provided with this paper.

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

## Acknowledgements

We thank Carina Lauter for her expert technical assistance. This work was supported by Macquarie University, Melanoma Institute Australia, the New South Wales Department of Health, NSW Health Pathology, National Health and Medical Research Council of Australia (NHMRC; grants 1093017, 1128951, 1130423, 2012860), Cancer Institute NSW and the Sydney Vital Translational Cancer Research Centre. R.A.S. is supported by an NHMRC Practitioner Fellowship (1141295). J.H.L. (2008735) and G.V.L. (2007839) are supported by NHMRC Fellowships. G.V.L. is also supported by the University of Sydney Medical Foundation. A.M.M. is supported by a Cancer Institute NSW Fellowship.

## Author contributions

S.Y.L., E.S., J.H.L., and H.R. designed the study and wrote the manuscript. S.Y.L., E.S., and H.R. analyzed the data. S.Y.L., E.S., B.P., A.S., Z.M., and M.I. conducted the experiments. J.H.L., B.S., R.P.S., A.M.M., M.S.C., R.A.S., and G.V.L. provided clinical data and input. All authors read, corrected, and approved the manuscript. S.Y.L. led the analysis of the PD1 PROG cell lines, which informed the analysis of the tumor dissociates, which was led by E.S. S.Y.L. prepared the initial manuscript draft focused on the cell line aspects of the study, and E.S. added the significant tumor dissociate analysis and conclusions. The order of the first authors, S.Y.L. and E.S., was agreed to by both authors and corresponding author H.R.

## Competing interests

M.S.C. is a consultant advisor to MSD, BMS, Novartis, Roche, Pierre Fabre, Sanofi, Merck Serono, Nektar, Eisia, and Ideaya and received honoraria from MSD, BMS, and Novartis. J.H.L. has received honoraria from AstraZeneca and travel support from BMS and Novartis. R.A.S. has received fees for professional services from F. Hoffmann-La Roche Ltd, Evaxion, Provectus Biopharmaceuticals Australia, Qbiotics, Novartis, Merck Sharp & Dohme, NeraCare, AMGEN Inc., Bristol-Myers Squibb, Myriad Genetics, and GlaxoSmithKline. G.V.L. is a consultant advisor for

Aduro Biotech Inc, Agenus Inc, Amgen Inc, Array Biopharma Inc, Boehringer Ingelheim International GmbH, Bristol-Myers Squibb, Evaxion Biotech A/S, Hexel AG, Highlight Therapeutics. S.L., Merck Sharpe & Dohme, Novartis Pharma AG, OncoSec, Pierre Fabre, QBiotics Group Limited, Regeneron Pharmaceuticals Inc, SkylineDX. B.V., Specialised Therapeutics Australia Pty Ltd. A.M.M. has participated in advisory boards for BMS, MSD, Novartis, Roche, and Pierre Fabre. The remaining authors declare no competing interests. R.P.M.S. has received honoraria for advisory board participation from MSD, Novartis, and Qbiotics and speaking honoraria from BMS and Novartis.
