## [Peer Review File · Nature Communications]

The molecular and functional landscape of resistance to immune checkpoint blockade in melanomaREVIEWERS' COMMENTS:

Reviewer #1 (Remarks to the Author): expertise in melanoma therapeutic response

The authors here present an excellent body of work that sheds much needed light on resistance mechanisms to ICB in melanoma. These findings will be of broad interest and importance to fields of melanoma, immunotherapy and immune evasion. Below are minor comments/suggestions.

- Would have been helpful if at least 1-2 short-term cultures from melanoma patients that responded well to ICB could have been concluded to serve as references to the short-term cultures from progressing tumors. I can understand feasibility being an issue, but this would have been excellent throughout all the figures of the manuscript.

Figure S2 B

- REVERT looks slightly misaligned

Table 1

- I really like this table. It answers a lot of confusion I had when reading the paper

Page 4

- B2M gene define
- Further describe the antigens associated with de-differentiation
- IFN γ signaling is said to also impact MHC expression. What was the original impact?
- "Several other features" – I am not sure what they are referring to (alterations? Markers of resistance?)
- The last paragraph is confusing. First the resistance features are unclear, then some resistance mechanisms are stated, but then more mechanisms are also reported. PD-L1 is included in both the unclear and reported aspects.

Page 5

- Define PD1 PROGs
- "and tumor-intrinsic IFN γ signaling" does not fit in the sentence grammatically to make sense. Is it part of the dedifferentiation or part of the immune cell exclusion?

Page 6

- How were they able to determine the difference between innate progressive disease and acquired resistance in 19 patients (Table S4)?
- Were the 10 patients that received prior systemic therapy part of the innate progressive disease group or the acquired resistance group?
- The cell lines that came from biopsies were differentiated into innate or acquired resistance groups, but that is after the patient study was done, not beforehand. My first question of Page 6 still stands.
- Sentence with AXL is a run-on sentence

Page 7

- Intrinsic IFN γ signaling is only found in 7 PD1 PROGs. How is this more common?
- What was the signature used to define melanocytic, transitory, neural crest-like and undifferentiated cell lines.
- Is undifferentiated the same as de-differentiated? Both phrases are used in the text.

Page 8

- Further explain the melanoma differentiation states
- "We also confirmed elevated protein expression of IRF1, PD-L1, PD-L2 and HLA-ABC in most of the PD1 PROGs with elevated baseline IFN γ activity (Fig. 2D, Fig. S2B)" What about these protein expressions for the 7 PD1 PROGs with intrinsic IFN γ signaling?

- Only half of the tumor biopsies showed this trend. And sample size of 6 seems small.

Page 9

- What was the percentage of T reg cells?
- The frequency of CD8 T cells could also be due to Treg activity, not necessarily just absence of antigens

Page 10

- Why were only 2 PD1 PROG cell lines looked into for MHC-1 expression?

Page 11

- Why is data not shown? Not even in supplementary?
- For the cell lines with homozygosity, was it for or against MHC-1/2 loci?

Page 12

- Treg activity should be looked into as well for MHC expression results
- Y366* does not have a reference for the asterisk anywhere on the page
- Still not sure how immune cell exclusion is common in these tumors...

Page 14

- There seems to be too few cell lines to say that the mechanism is important in melanoma
o De-differentiation (1/6)
o Loss of B2M (1/6)

Page 36

- Figure 2B seems out of place. What is its purpose? How are these scores decided?
- Figure 2C AXL has a stark difference in the cell lines on the right. Why are they not with the differentiated melanoma cells on the left?

Page 42

- Figure 5A is not labeled well. It is hard to tell what parts of the graph the top two cell lines are referring to
- Figure 5F – the bottom left flow cytometry analysis looks too randomly gated

Page 44

- Figure 6A is not labeled well. It is hard to tell what parts of the graph the cell lines are referring to
- Figure 6D – the selected cell lines should be labeled a different color to highlight which points they are referring to

Page 45

- Caption for Figure 6H is not fully described and is part of the caption for Figure 6G

Page 46

- Figure 7B needs to specify which cell lines these alterations are in (JAK2, B2M, LOH)

Reviewer #2 (Remarks to the Author): expertise in antigen presentation and immunology

The manuscript by Lim et al. describes molecular changes in cohorts of melanoma patients showing resistance to checkpoint inhibitors.

Strengths – The manuscript is well written and often demonstrates observations using multiple different techniques. The datasets are generally well controlled and appropriately analysed.

Weaknesses – Some conclusions are derived from low numbers of tumours (e.g. SCC16-0016 cell line with disrupted IFN-g signalling) and it is unclear if many of the reported properties might be seen in melanoma tumours/lines that have not undergone checkpoint blockade. If so, then it is

difficult to conclude that the different programming is as a result of resistance to checkpoint therapy. The initial, proposed comparison between innate vs acquired checkpoint progression patients did not extend strongly through the analysis of the data. However, as a catalog of changes seen in patients who happen to be on checkpoint therapy, the dataset is informative.

Specific points to improve the manuscript include :

- (1) In Fig. 1B, is the SCC16-0016 cell line showing resistance to IFN-g in expression of JAK2 transcript? Although starting from a lower baseline transcript expression (control), is the increase with IFN-g any less than some of the responding cell lines that start at a higher baseline level of transcript?
- (2) The use of 1000 U/mL IFN-g seems excessive (any changes to cell viability?). Was this titrated to determine an optimal concentration?
- (3) Fig. 2E + F – it was unclear why some of the cytokines induced in the absence of IFN-g (Fig. 2E) in the intrinsic lines were apparently not induced (inhibited?) by IFN-g in those same cell lines when IFN-g is added (Fig. 2F). Is Fig. 2F, the fold change over no addition of IFN-g or something else? Figures need further explanation.
- (4) Fig. 3A left panel – what does the dotted line represent?
- (5) Fig. 3C – while it is interesting to dissect the cell populations from tumour which is unresponsive to IFN-g, it is difficult to make solid conclusions from n=1.
- (6) Fig. 4B – have the WMD-084#2 and WMD-084#1 cell lines been profiled for MHC I expression? It is unusual for a high concentration of peptide (10ug/mL) to not increase CD8 T cell activity from endogenous peptide levels in the #1 line and then for this saturating peptide addition to not increase the CD8 T cell activity level in the #2 line to the level in #1 line. One explanation might be a difference in MHC I level between the melanoma lines.
- (7) Fig. 7A right panel – this is not a very convincing “restoration” of CD8 T cell recognition by transfecting with HLA-A2. Were successfully transfected cells sorted and how much HLA-A2 was expressed on their surface?

Reviewer #3 (Remarks to the Author): clinical expertise in melanoma

In Lim et al, the authors developed short term tumor cell lines and matched tumor samples from melanoma patients progressing on immune checkpoint inhibitor therapy. Several molecular programs are found to be correlated with resistance, including IFN γ signaling, melanoma de-differentiation, MHC, and PTEN loss.

The topic of resistance to ICI is of course, important, given its frequent use of this form of treatment in cancer broadly and in melanoma specifically. The generation of these cell lines, particularly matched cell lines from the same patient, is an important resource. Nevertheless, the data provided is mostly correlative and the pathways identified are not novel. These factors reduce the enthusiasm I have for this manuscript.

A more complete evaluation of the drivers of resistance, as well as a more unbiased evaluation of the data, would be helpful. In my opinion, these limitations preclude publication at Nature Communications, at least in the current form. Significantly more analysis and molecular studies would be required.

There is relatively scant evidence for most of the claims made in the manuscript that molecular changes are driving resistance. For example, the authors state that “IFN γ signaling was disrupted in the SCC16-0016 PD1 PROG cell line because of a genomic deletion/fusion event involving the JAK2 and INSL6 genes on chromosome band 9p24.1.” However, how do the authors know that the loss of JAK2 is compromising IFN γ signaling? This could be ascertained by re-introducing JAK2 into these cells. Presumably, there are other mutations observed in the exome analysis of this patient sample?

Similarly, the statement that “Melanoma-intrinsic IFN γ signaling is immune suppressive” is supported by gene expression of IFN γ regulated genes and an inflammatory secretome and some multiparameter flow cytometry of defined immune attributes.

The concept that de-differentiation of melanomas drives resistance through loss of antigen repertoire is supported through the use of pulsing of the Melan-A peptide leading to an increase in activated CD8+ T cells.

The effect of PTEN loss on some immune cell infiltration is merely correlative. The observation that PTEN loss is seen in several innately resistant brain metastasis could be a reflection that PTEN loss contributes to brain metastasis rather than a mechanism of resistance.

RESPONSE TO REVIEWERS' COMMENTS

Reviewer #1

- 1) At least 1-2 short-term cultures from melanoma patients that responded well to ICB could have been concluded to serve as references to the short-term cultures from progressing tumors.

We have now added seven short-term melanoma cell models to this manuscript. These new cell models were derived from patients who were not on systemic therapy (PRE) and two of these PRE cell lines (SMU16-0570 and SCC16-0040) came from patients who went on to receive and respond to ipilimumab and nivolumab (see new Table S5).

These PRE cell models were analysed for key immune checkpoint inhibitor resistance mechanisms detailed in this paper. We have summarised the new data in Supplementary Figure 8 and included the details in the results section, page 13, as follows:

“We also examined whether immune checkpoint inhibitor mechanisms were enriched or selected in short-term melanoma cell models derived from patients prior to treatment with systemic therapy. We identified seven pre-treatment melanoma cell lines (PRE), and two of these (SMU16-0570 and SCC16-0040) came from patients who subsequently went on to receive and respond to combination ipilimumab and nivolumab (Table S5). All seven PRE melanoma cell lines were heterozygous across the MHC-I locus (Data S5), all accumulated cell surface MHC-I and MHC-II, and all responded to exogenous IFN γ by inducing cell surface expression of the MHC molecules (Fig. S8). Based on the baseline expression of IFN γ -regulated targets PD-L1 and PD-L2, only the SMU16-0570 PRE cell line showed elevated PD-L1 and PD-L2 expression, indicative of intrinsic IFN γ activity (Fig. S8). Predictably, SMU16-0570 was the only PRE melanoma cell line that displayed consistent features of de-differentiation, that included the accumulation of AXL and concurrent loss of MITF, SOX10 and Melan-A (Fig. S8). Finally, the SMU16-0570 PRE melanoma, which was derived from a brain metastasis, had an inactivating PTEN mutation (p.A192fs*20). Missense PTEN mutations were identified in the SCC16-0323 and WMD-031 PRE (Table S5), although these mutations are of unknown clinical significance³⁸.”

- 2) Figure S2B, REVERT looks slightly misaligned

The REVERT image has now been adjusted so that it better aligns with each of the IRF sample lanes.

- 3) Page 4: B2M gene define.

B2M has now been defined as beta-2-microglobulin.

- 4) Page 4: Further describe the antigens associated with de-differentiation.

We have now provided some key examples of melanosomal differentiation antigens,

“...silencing of melanosomal wild-type differentiation antigens (e.g MART-1/Melan-A, gp100, TYR) during melanoma de-differentiation...”

- 5) Page 4: IFN γ signaling is said to also impact MHC expression. What was the original impact?

We have clarified that defects in IFN γ signaling would prevent IFN γ -induced MHC expression.

“Defects in IFN γ signaling, which also prevent IFN γ -induced MHC expression, have been identified in 4-10% of PD1 resistant melanomas and are often due to loss-of-function mutations in the IFN γ effector kinases JAK1 and JAK2^{9,16}.”

- 6) Page 4: “Several other features” – I am not sure what they are referring to (alterations? Markers of resistance?).

This sentence has been updated to the following:

“Recent studies have identified features within the tumor microenvironment and the gut microbiome that are associated with immune checkpoint inhibitor resistance, but the precise contribution of these characteristics remain unclear.”

- 7) Page 4: The last paragraph is confusing. First the resistance features are unclear, then some resistance mechanisms are stated, but then more mechanisms are also reported. PD-L1 is included in both the unclear and reported aspects.

We have reworded this paragraph to improve clarity.

“For instance, poor responses to immune checkpoint inhibitors are associated with limited gut microbiome diversity and gut inflammation¹⁷⁻²², induced expression of immune checkpoint molecules (including PD-L1, LAG-3, TIM-3)^{23-25,26,27}, and immune cell exclusion within the tumor microenvironment²⁸. Mechanisms driving some of these putative markers of resistance have also been reported. For instance, induction of immune inhibitory ligands such as PD-L1, LGALS9, TNFRSF14 may reflect an adaptive negative feedback mechanism involving sustained IFN γ signaling (rather than loss of IFN γ activity)²⁵ while CD8 T cell exclusion from the tumor microenvironment may be caused by an immunosuppressive tumor cell secretome driven by aberrant β -catenin, PTEN loss or CDK-cell cycle signaling pathways²⁹⁻³¹.”

- 8) Page 5: Define PD1 PROGs

We have clarified the definition of PD1 PROGs in this section and, again, in the results section, page 6 paragraph 1.

“In this study, we functionally dissected tumor-intrinsic mechanisms of immunotherapy resistance in a unique panel of short-term melanoma cell lines, termed PD1 PROGs, and matched tumor biopsies derived from 18 patients progressing on PD1 inhibitors, either alone or in combination with ipilimumab.”

- 9) Page 5: “and tumor-intrinsic IFN γ signaling” does not fit in the sentence grammatically to make sense. Is it part of the dedifferentiation or part of the immune cell exclusion?

We have updated the sentence to improve clarity.

“These programs include i) the concurrent genetic and epigenetic disruption of the MHC proteins, ii) diminished expression of wild-type antigens via de-differentiation and tumor-intrinsic IFN γ signaling and iii) immune cell exclusion associated with PTEN loss and brain metastasis.”

- 10) Page 6: How were they able to determine the difference between innate progressive disease and acquired resistance in 19 patients (Table S4)?

The criteria used to define innate vs acquired progressing lesions are described in the Results Section, Page 6, last paragraph. We have reproduced the definition here for clarity:

“Innate progressing lesions were defined as an increase in size of pre-existing metastases that never underwent tumor shrinkage or new metastases identified within 6 months of starting treatment. Acquired progressing lesions were defined as pre-existing tumors that initially underwent tumor shrinkage by >30% from baseline but subsequently progressed on PD1 inhibitor or new metastases identified after 6 months of starting PD1 inhibitor.”

- 11) Page 6: Were the 10 patients that received prior systemic therapy part of the innate progressive disease group or the acquired resistance group?

These details are provided in Table S4. We have included an additional sentence to highlight the fact that innate progressive PD1 PROGs were predominantly derived from patients who had received prior therapy. Page 6, last sentence:

"It is worth noting that 10/13 innate PD1 PROGs were derived from eight patients who received prior systemic therapy (Table S4)."

- 12) Page 6: The cell lines that came from biopsies were differentiated into innate or acquired resistance groups, but that is after the patient study was done, not beforehand. My first question of Page 6 still stands. Please refer to Reviewer #1, point 10

- 13) Page 6: Sentence with AXL is a run-on sentence
We could not find the run-on sentence referred to here.

- 14) Page 7: Intrinsic IFN γ signaling is only found in 7 PD1 PROGs. How is this more common?
The results section heading has been updated to more accurately reflect the data.

"Intrinsic IFN γ signaling is more common than loss of IFN γ activity in immune checkpoint inhibitor resistant melanoma".

- 15) Page 8: What was the signature used to define melanocytic, transitory, neural crest-like and undifferentiated cell lines. Further explain the melanoma differentiation states

The signatures used to define the four melanoma cell states (undifferentiated, neural crest-like, transitory and melanocytic) were published in (Tsoi et al. Cancer Cell 33:890-904). We have updated the methods to include the precise details of these analyses. We have also clarified that these transcriptome subtypes correspond to the melanocytic (transitory and melanocytic) and dedifferentiated melanoma (undifferentiated, neural crest-like) phenotypes, page 19, paragraph 2:

"To obtain abundance values corrected for transcript lengths as required by the single-sample gene set enrichment analysis (ssGSEA; ⁶³), RSEM was used to derive the FPKM estimates using GENCODE Genes version 26 as the reference transcript database. ssGSEA was used to derive the absolute enrichment scores using the gene sets from the Molecular Signature Database version 6.2 ⁶⁴. The same FPKM values were used to determine the melanoma differentiation transcriptome subtypes (undifferentiated, neural crest-like, transitory and melanocytic) using the support vector machine "top-scoring pairs" scripts kindly provided by Dr T. Graeber³⁴. These transcriptome subtypes correspond to the melanocytic (transitory and melanocytic) and dedifferentiated (undifferentiated, neural crest-like) melanoma phenotypes^{65,66}."

- 16) Page 8: "We also confirmed elevated protein expression of IRF1, PD-L1, PD-L2 and HLA-ABC in most of the PD1 PROGs with elevated baseline IFN γ activity (Fig. 2D, Fig. S2B)". What about the protein expressions for the 7 PD1 PROGs with intrinsic IFN γ signaling?

The protein analyses include the 7 PD1 PROGs with intrinsic IFN γ signaling. We have updated this sentence and the Figure 2D figure legend to clarify this point on page 8, Paragraph 3:

"We also confirmed that most of the six PD1 PROGs with elevated baseline IFN γ activity accumulated high levels of IRF1, PD-L1, PD-L2 and MHC-I (Fig. 2D, Fig. S2B).

"Figure 2D Legend: Plots showing IRF1 protein expression (derived from the densitometric normalized protein data after log₂ transformation and z score calculation), relative cell surface expression (median fluorescence intensity stained divided by fluorescence minus one control, MFI/FMO) of PD-L1, PD-L2 and MHC-I in six PD1 PROGs with intrinsic IFN γ activity compared to 15 PD1 PROGs without intrinsic IFN γ activity. Data compared using Mann-Whitney test."

- 17) Page 8: Only half of the tumor biopsies showed this trend. And sample size of 6 seems small.

We agree that the sample size is small, but the findings are particularly interesting, and we have been very careful to highlight the limited sample size and trend in the data:

“Although the numbers are small, we noted that PD1 PROG cells with intrinsic IFN γ activity were derived from tumor biopsies with a trend towards lower CD45 $^+$ cell content (3/5 tumor dissociates with < median (12.3%) CD45 $^+$ cells) and high macrophage content (2/5 dissociates with > top quartile (27%) macrophages) (Fig. 3A, Table 1).”

We also note that the data in Figure 3B are compelling, showing strong inverse correlations between IFN γ signaling and CD8 $^+$ T cell populations in melanoma tissue.

- 18) Page 9: What was the percentage of T reg cells? The frequency of CD8 T cells could also be due to Treg activity, not necessarily just absence of antigens

We have updated the analysis in this section and now show in a new Figure 3C that Treg frequency is not correlated with activated CD8 T cell frequency.

“The diminished frequency of activated CD8 $^+$ T cell subsets did not correlate with the presence of regulatory (CD4 $^+$ FOXP3 $^+$) T cells in PD1 PROG tumors, irrespective of the IFN γ intrinsic activity (Fig. 3C).

The frequency and activation markers of Tregs in the SCC16-0016 sample has also been updated to improve clarity on page 9, paragraph 2:

“...NK cells may be suppressed by the presence of highly activated regulatory (CD4 $^+$ FOXP3 $^+$) T cells³⁵ (8.1% of CD45 $^+$ cells were regulatory T cells and the CD8/TReg ratio = 1.8 was in the bottom quartile; Fig. 3D). The Treg cells in the SCC16-0016 tumor dissociate expressed markers indicative of functional activation, with >80% of Treg cells expressing the activation markers CD38, ICOS and OX40 (Fig. 3D).

- 19) Page 10: Why were only 2 PD1 PROG cell lines looked into for MHC-1 expression?

The expression of MHC-I and MHC-II was examined in all 22 PD1 PROG cell lines (see Figures 5A and 6A). Alterations affecting MHC-I and/or MHC-II were identified in 6 PD1 PROG cell lines, and these alterations were confirmed and validated in the corresponding cell lines and, where available, in the matched tumour dissociates. For instance, the data for B2M mutation is summarised in Figures 2A-E. The data describing CIITA silencing is shown in Figures 3A-D. Loss of MHC-I/II heterozygosity is tabulated in Supplementary Data 5.

- 20) Page 11: Why is data not shown? Not even in supplementary?

The RNA sequence data is available and CIITA transcript sequence can be assessed from the available RNA sequence data (PRJNA818797). The promoter sequences can be requested - but as these are capillary sequences and all were wild type, there was not much value in including as supplementary data.

- 21) Page 11: For the cell lines with homozygosity, was it for or against MHC-1/2 loci?

Homozygosity was only seen in HLA-A, and the complete HLA typing data is shown in Supplementary Data 5.

- 22) Page 12: Treg activity should be looked into as well for MHC expression results

We have re-examined the Treg activity in relation to MHC-I/II expression in 19 tumour dissociates. The data are now shown in the updated Figure 7C, and confirms that in melanoma with low (below median) MHC-I and/or MHC-II expression there are fewer CD45 $^+$, CD8 $^+$ and T $\alpha\beta$ cells, with no detectable differences in regulatory T cells. The following details have been included in page 12:

“In melanoma with low MHC-I/II expression (defined here as < median expression), there were fewer CD45⁺, CD8⁺ and TCR $\alpha\beta$ cells in the tumor microenvironment (Fig. 7C, Table 1). The frequency of CD4⁺ and regulatory T cells and macrophages did not differ according to MHC-I/II melanoma expression (Fig. 7C).”

23) Page 12: Y366* does not have a reference for the asterisk anywhere on the page.

The asterisk in Y366* is a standard nomenclature for a termination codon, and we have indicated that the Y366* is a terminating mutation.

“... SCC15-0528 had a homozygous terminating Y336* mutation (c. 1008C>G).”

24) Page 12: Still not sure how immune cell exclusion is common in these tumors.

We have updated the title of this section to more accurately reflect the data.

“Diminished immune cell infiltration is associated with immune checkpoint inhibitor resistant brain tumors”

25) Page 14: here seems to be too few cell lines to say that the mechanism is important in melanoma: De-differentiation (1/6). Loss of B2M (1/6)

The statement referred to by the reviewer (now on page 15) is not indicating that B2M loss or dedifferentiation only occurs in 1/6 PD1 PROGs, but rather that homozygosity of HLA-A alleles co-exists with other resistance effectors including PTEN loss (4/6), B2M loss (1/6) and de-differentiation (1/6).

26) Figure 2B: seems out of place. What is its purpose? How are these scores decided?

Figure 2B confirms that the six PD1 PROG cells with intrinsic IFN γ activity are:

- i) enriched for transcriptome signatures associated with invasion and mesenchymal transition and
- ii) show downregulation of proliferative transcriptome signatures.

These details are described in page 7, paragraph 2:

“Geneset enrichment analysis confirmed enrichment of Hallmark interferon transcriptome gene sets, in the absence of IFN γ stimulation, in these six cell lines, along with strong enrichment of mesenchymal and invasive signatures. This was accompanied by the loss of Hallmark proliferative transcriptome signatures including the estrogen response and oxidative phosphorylation gene sets (Fig. 2B, Data S1).”

27) Figure 2C: AXL has a stark difference in the cell lines on the right. Why are they not with the differentiated melanoma cells on the left?

The classification of PD1 PROG melanomas were based on the combination of well-established differentiation markers including MITF, SOX10, Melan-A and AXL. Taking these markers together the cells on the right were more closely related to dedifferentiated melanomas (based on unbiased Euclidean clustering). This was validated with the transcriptome-based differentiation signatures with no melanocytic cells in the de-differentiated cluster. It is also worth noting that the cells on the right had low, but detectable AXL expression.

28) Figure 5A and 6A: not labelled well. It is hard to tell what parts of the graph the top two cell lines are referring to

We have modified all graphs and figure legends throughout the manuscript to highlight data points that are labelled.

29) Figure 5F: the bottom left flow cytometry analysis looks too randomly gated

We have updated Figure 5F to include the T cells only control which was used to derive the gating strategy for the co-culture experiments. The figure legend has also been updated to clarify the gating strategy details:

“The expression of CD107 and IFN γ in T cell monocultures (left panels) was used to establish the gating strategy for these experiments.”

- 30) Figure 6D: the selected cell lines should be labelled a different color to highlight which points they are referring to

We have modified all graphs and figure legends throughout the manuscript to highlight data points that are labelled.

- 31) Figure 6H: not fully described and is part of the caption for Figure 6G

We have separated the figure legends for Figure 6H and 6G as shown below:

“6G) Representative histograms showing melanoma MHC-II expression in MHC-II^{low} SCC17-0263 and SCC11-0270 PD1 PROGs treated with 1000 U/ml IFN γ (shaded red histograms) or IFN γ with panobinostat (HDACi; shaded blue histograms).

6H) Bar graphs show relative MHC-II expression (IFN γ /control-treated) in MHC-II^{low} SCC17-0263 and SCC11-0270 PD1 PROGs treated with 1000 U/ml IFN γ or IFN γ with panobinostat (HDACi). Individual values and mean of three biological replicates \pm sd are shown and paired, two-tailed T-test was used to compare the data.”

- 32) Figure 7B: needs to specify which cell lines these alterations are in (JAK2, B2M, LOH).

We have included in the figure legend the cell lines with the specific alterations and also highlighted in the figure. The updated figure 7B legend is shown:

“7B: Scatterplot showing correlation of MHC-II and MHC-I expression score (MFI melanoma/MFI TILs) in PD1 PROG tumor dissociates (n=20). Correlation calculated using the Spearman's rank correlation coefficient. Tumor samples with established alterations in JAK2 (SCC16-0016), MHC-I/II (WMD17-0112, SMU-059), B2M/CIITA (SMU-092, SCC13-0156), are highlighted and color coded. Three tumors with low MHC-I/II expression on melanoma without causal mechanisms are circled. Dotted lines indicate median MHC-I and MHC-II expression scores.”

Reviewer #2

- 1) Some conclusions are derived from low numbers of tumours (e.g. SCC16-0016 cell line with disrupted IFN-g signalling) and it is unclear if many of the reported properties might be seen in melanoma tumours/lines that have not undergone checkpoint blockade. If so, then it is difficult to conclude that the different programming is as a result of resistance to checkpoint therapy.

We appreciate the comments regarding selection of mutations post immune checkpoint inhibitor treatment. We have now included a new results section (page 13: **Immune checkpoint inhibitor resistance mechanisms are not enriched in PRE-treatment melanoma**) examining seven short-term melanoma cells lines derived from patient not on systemic treatment. Importantly we confirm that the alterations detected in immune checkpoint resistant PD1 PROGs were not enriched in these pre-treatment melanomas. Please see detailed response to Reviewer 1, comment 1.

- 2) The use of 1000 U/mL IFN-g seems excessive (any changes to cell viability?). Was this titrated to determine an optimal concentration?

We selected a relatively high dose of IFN γ to allow for the analysis of the IFN γ effects on gene expression and cell cycle distribution. In order to measure IFN γ -induced cell cycle changes we selected 1000 U/ml IFN γ based on the study by Kortylewski et. al. (J Invest Dermatol 122:414) - these cell cycle data are being prepared for a separate study.

Importantly, we confirmed that induction of IFN γ targets MHC-I and MHC-II was not affected by increasing the concentration of IFN γ (see Figure below).

IFN γ -concentration effects on the expression of MHC-I and MHC-II on the indicated PD1 PROG melanoma cells. The FMO (fluorescence minus one) control shows baseline fluorescence in the absence of MHC antibody. The IFN γ concentration used (units/ml; U/ml) is shown and the grey line marks baseline MHC expression in the absence of exogenous IFN γ .

We have also included the following sentence in the methods section of the manuscript:

“The concentration of IFN γ was based on ⁶¹ and induced maximal levels of MHC-I and MHC-II in titration experiments (data not shown).”

- 3) Fig. 1B: is the SCC16-0016 cell line showing resistance to IFN-g in expression of JAK2 transcript? Although starting from a lower baseline transcript expression (control), is the increase with IFN-g any less than some of the responding cell lines that start at a higher baseline level of transcript?

There is a slight increase in JAK2 transcript levels post IFN γ in the SCC16-0016 cell line (Figure 1B: log₂ fold change post IFN γ /pre IFN γ = 0.6). For the reviewer's interest, this level of induction was the second lowest amongst the 22 PD1 PROG cell lines (WMD17-0112 had a log₂ fold change = 0.41, with JAK2 post IFN γ expression = 4.18).

It is important to highlight the limitations of relying on this single data point, i.e. the next generation RNA sequencing data shown in Figure 1B is based on a single RNA sequencing experiment. Most importantly, for the SCC16-0016 cell line, Figure 1C confirms no JAK2 protein pre and post IFN γ treatment; Figure 1D shows no IFN γ -induction of the IFN γ targets MHC-I, MHC-II, PD-L1 and PD-L2; and Figures 1E-F, shows

no proliferation effects in response to IFN γ . Figures 1C-F are based on at least three independent biological replicates.

- 4) Fig. 2E + F – it was unclear why some of the cytokines induced in the absence of IFN-g (Fig. 2E) in the intrinsic lines were apparently not induced (inhibited?) by IFN-g in those same cell lines when IFN-g is added (Fig. 2F). Is Fig. 2F, the fold change over no addition of IFN-g or something else? Figures need further explanation.

We have revised the figure legends to clarify the details of these data and updated the figures to more clearly describe the log₂ fold change data.

With regard to the query that many cytokines overexpressed in the IFN γ -high compared to the IFN γ -low PD1 PROGs - we described these results on page 8, paragraph 2:

“Most secreted factors (TNF α , GM-CSF, G-CSF, PDGF-BB, CCL5, CCL11, IL-3, IL-6) were not induced by exogenous IFN γ in our panel of melanoma cells (Data S4; Fig. 2F), confirming the complex inflammatory signaling profile of de-differentiated PD1 PROGs with intrinsic IFN γ signaling.”

- 5) Fig. 3A left panel – what does the dotted line represent?

We have updated this figure and set the vertical dotted line to 12.3% of total, as described in the text (page 8, last paragraph) and figure legend:

Figure 3A: “...The dotted vertical line indicates the median percentage of CD45⁺ cells (12.2% of total cells).”

- 6) Fig. 3C – while it is interesting to dissect the cell populations from tumour which is unresponsive to IFN-g, it is difficult to make solid conclusions from n=1.

We agree with the reviewer, and have highlighted that we only identified a single PD1 PROG with loss of IFN γ signalling. As such, we have revised the paragraph to:

“Although, we identified only one PD1 PROG with loss of IFN γ signaling, it was interesting to note that this tumor had low frequency of CD8⁺ T cells (14.4% of CD45⁺ cells; bottom quartile <26%), and the highest percentage of natural killer (NK) cells in our tumor dissociate panels (11% of the CD45⁺ cells) (Fig. 3C).”

- 7) Fig. 4B – have the WMD-084#2 and WMD-084#1 cell lines been profiled for MHC I expression? It is unusual for a high concentration of peptide (10ug/mL) to not increase CD8 T cell activity from endogenous peptide levels in the #1 line and then for this saturating peptide addition to not increase the CD8 T cell activity level in the #2 line to the level in #1 line. One explanation might be a difference in MHC I level between the melanoma lines.

Both WMD-084#1 and WMD-084#2 have been profiled for MHC-I expression, and the baseline and IFN γ -induced expression is tabulated below (for the reviewer’s information).

	Control-treated MHC-I (MFI/FMO)	IFNγ-treated MHC-I (MFI/FMO)
WMD-084#1	9 \pm 0.4	26 \pm 1
WMD-084#2	9 \pm 1.7	24 \pm 4.9

MFI, Median fluorescence intensity; FMO, fluorescence minus one control. Median \pm standard deviation shown.

We have included these data in the new Figure 4C panel with updated figure legend:

Figure 4C “Cell surface expression (median fluorescence intensity divided by fluorescence minus one control, MFI/FMO) of MHC-I in the indicated cell lines at 72h after treatment with BSA control or IFN γ (1000 U/ml). Average of 3 biological replicates shown for each cell line.”

We have also updated Figure 4B, right panel to show the difference (rather than fold change) of double positive activated T cells – based on the revised method described below, reviewer #2, point 8.

- 8) Fig. 7A right panel – this is not a very convincing “restoration” of CD8 T cell recognition by transfecting with HLA-A2. Were successfully transfected cells sorted and how much HLA-A2 was expressed on their surface?

We have repeated the HLA restoration experiments with sorted HLA-A2 positive transfected melanoma cells. We now show that 97% of WMD17-0012 PD1 PROG cells displayed HLA-A2 expression post sorting; new Figure 7A).

Analysis of double positive IFN γ /CD107 reactive T cells confirmed that the magnitude of response is low in the presence of transfected HLA-A2 and Melan-A peptide, but significantly elevated (new Figure 7A middle and right panels).

These data are consistent with previous reports using TILs derived from immune checkpoint inhibitor resistant melanoma (see Andersen et al. 2018. *Ann Oncol.* 29:1575-1581). Andersen et al, reported variable T cell responses ranging from 1% to 84%, and they used clear criteria for determining a positive anti-tumor response. These criteria have now been included in our methods as shown below:

Page 22: “For Melan A peptide loading, 1×10^5 melanoma cells were pulsed with 10 $\mu\text{g/ml}$ Melan A peptide (AAGIGILTV, Auspep, Australia) or DMSO for 1.5 h and washed before co-culture with TILs. Determination of a positive anti-tumor responses in co-culture experiments was based on criteria defined by 71 , and included a difference of $>0.1\%$ for double positive CD107/IFN γ T cells from the background (i.e co-culture experiments with untransfected or unpulsed cells).”

Our results also clarify the low reactivity seen after HLA-A2 restoration:

Page 11, second last paragraph: “Although the magnitude of response was low, there was a significant increase in double IFN γ /CD107 positive reactive T cells when HLA-A2:01 was restored in the WMD17-0112 cells (mean increase 2.7%; Fig. 7A).”

Reviewer #3

- 1) There is relatively scant evidence for most of the claims made in the manuscript that molecular changes are driving resistance. For example, the authors state that “IFN γ signaling was disrupted in the SCC16-0016 PD1 PROG cell line because of a genomic deletion/fusion event involving the JAK2 and INSL6 genes on chromosome band 9p24.1.” However, how do the authors know that the loss of JAK2 is compromising IFN γ signaling? This could be ascertained by re-introducing JAK2 into these cells. Presumably, there are other mutations observed in the exome analysis of this patient sample?

We respectfully disagree with reviewer’s comment. We validated the contribution of differentiation antigens (Figure 4B), the importance of B2M and MHC-I for T-cell recognition (Figure 5F) and the impact of HLA-A2 homozygosity (Figure 7).

In addition, as requested by this reviewer, we have now reintroduced wild-type JAK2 expression in the JAK2-mutant SCC16-0016 cell line. The data shown in Fig. S1D confirm that the introduction of wild-type JAK2 restored IFN γ -mediated MHC-I induction. These details are now included in page 7:

“The transient reintroduction of FLAG-tagged wild-type JAK2 into the SCC16-0016 PD1 PROG cell line restored IFN γ -mediated induction of MHC-I (Fig. S1).”

- 2) Similarly, the statement that “Melanoma-intrinsic IFN γ signaling is immune suppressive” is supported by gene expression of IFN γ regulated genes and an inflammatory secretome and some multiparameter flow cytometry of defined immune attributes. The concept that de-differentiation of melanomas drives resistance through loss of antigen repertoire is supported through the use of pulsing of the Melan-A peptide leading to an increase in activated CD8+ T cells.

We appreciate the reviewer’s feedback.

- 3) The effect of PTEN loss on some immune cell infiltration is merely correlative. The observation that PTEN loss is seen in several innately resistant brain metastasis could be a reflection that PTEN loss contributes to brain metastasis rather than a mechanism of resistance.

We agree with the reviewer’s comment - and have been careful throughout the manuscript to use the word association with regards to PTEN loss. For instance, in the abstract, we state that ‘immune cell exclusion is associated with loss of the PTEN tumor suppressor’, and in the discussion on page 14, we state ‘a smaller subset of PD1 inhibitor resistant melanomas (5/23; 22%) show loss-of-function mutations in the *PTEN* gene and this was associated with a paucity of immune cells and brain metastases’.

REVIEWERS' COMMENTS

Reviewer #2 (Remarks to the Author):

The authors have addressed the majority of my concerns in the revised paper. The results for 084#1 cell line in Fig. 4B still seems unusual and suggests that either this cell line is activating the CD8 T cells in a non-antigen specific manner or the endogenous production of peptide/MHC is saturating (this would seem unlikely). It is also noted that the flow cytometry plots provided for 084#1 do not appear to be representative of the summarised data in the graph (only n=3 in the graph but two of the results suggest that control activation > peptide activation and the other result does not have a difference of 1.6 as shown by the "representative" plots). Also the figure legend describes "fold change" but the figure looks like the difference is plotted. Authors need to clarify.

Reviewer #3 (Remarks to the Author):

The authors have significantly enhanced the manuscript addressing many comments by the reviewers.

With respect to comments by Reviewer #1: These comments have been adequately addressed by the authors.

With respect to comments by Reviewer #3: These comments have been adequately addressed by the authors.

RESPONSE TO REVIEWERS' COMMENTS

Reviewer #2 (comments to the author)

The authors have addressed the majority of my concerns in the revised paper. The results for 084#1 cell line in Fig. 4B still seems unusual and suggests that either this cell line is activating the CD8 T cells in a non-antigen specific manner or the endogenous production of peptide/MHC is saturating (this would seem unlikely). It is also noted that the flow cytometry plots provided for 084#1 do not appear to be representative of the summarised data in the graph (only n=3 in the graph but two of the results suggest that control activation > peptide activation and the other result does not have a difference of 1.6 as shown by the "representative" plots). Also the figure legend describes "fold change" but the figure looks like the difference is plotted. Authors need to clarify.

We have updated Figure 4B ensuring that the histogram matched the represented flow cytometry image. The figure legend has also been corrected as follows:

Bar graph shows mean \pm sd difference (Melan-A peptide pulsed minus control pulsed) in percentage of CD107⁺/IFN γ ⁺ CD8⁺ T cells (n=3 for SMU14-0301 and WMD-084#1 and n=4 for WMD-084#2 biologically independent experiments).